# A parameter-free statistical test for neuronal responsiveness

Jorrit S Montijn*, Koen Seignette, Marcus H Howlett, J Leonie Cazemier, Maarten Kamermans, Christiaan N Levelt, J Alexander Heimel*

Netherlands Institute for Neuroscience, Royal Netherlands Academy of Arts and Sciences, Amsterdam, Netherlands

**Abstract** Neurophysiological studies depend on a reliable quantification of whether and when a neuron responds to stimulation. Simple methods to determine responsiveness require arbitrary parameter choices, such as binning size, while more advanced model-based methods require fitting and hyperparameter tuning. These parameter choices can change the results, which invites bad statistical practice and reduces the replicability. New recording techniques that yield increasingly large numbers of cells would benefit from a test for cell-inclusion that requires no manual curation. Here, we present the parameter-free ZETA-test, which outperforms t-tests, ANOVAs, and renewal-process-based methods by including more cells at a similar false-positive rate. We show that our procedure works across brain regions and recording techniques, including calcium imaging and Neuropixels data. Furthermore, in illustration of the method, we show in mouse visual cortex that (1) visuomotor-mismatch and spatial location are encoded by different neuronal subpopulations and (2) optogenetic stimulation of VIP cells leads to early inhibition and subsequent disinhibition.

**\*For correspondence:**
jsmontijn@gmail.com (JSM);
a.heimel@nin.knaw.nl (JAH)

**Competing interest:** The authors declare that no competing interests exist.

## Introduction

Many neuroscience studies rely on the analysis and visualization of neuronal spiking signals. Classical studies used manual curation during experiments to select cells for analysis (*Hubel and Wiesel, 1959*; *Mountcastle, 1957*), but this method cannot provide a statistically unbiased sample. Moreover, such manual curation is unsuitable for state-of-the-art large-scale recording techniques, such as Neuropixels and high-density multi-electrode arrays (*Bartolo et al., 2020*; *Jun et al., 2017*; *Semedo et al., 2019*; *Steinmetz et al., 2019*).

Despite the widespread application of neuronal responsiveness analyses, neuroscience currently lacks a standard practice for determining whether a neuron is responsive to an experimental stimulus or treatment (*Mesa et al., 2021*). Common approaches, such as comparing a neuron's average spike rate during the presence and absence of a stimulus, can only detect mean-rate modulated cells (*Mazurek et al., 2014*; *Ringach et al., 2002*). On the other hand, approaches such as computing a peri-stimulus time histogram (PSTH) and applying an ANOVA, require the a priori selection of an arbitrary binning window size (*Palm et al., 1988*). Choosing the wrong bin size reduces the test's sensitivity, while optimizing from a range of window sizes creates a multiple-comparison problem. This lowers the approach's statistical power when corrections are applied, or can even lead to (unintentional) 'p-hacking' if the results are not corrected (*Head et al., 2015*). Finally, while (point-process) model-based approaches can circumvent many of the above problems (*Kass et al., 2014*), they still require the a priori selection, or tuning, of hyperparameters specific to the statistical properties of classes of cells, or even individual neurons. Many model-based approaches are therefore not well suited to an unsupervised analysis of large-scale data recorded with state-of-the-art techniques.

To solve these problems, we developed a method that detects whether a cell is responsive to stimulation in a statistically robust way and avoids binning and parameter selection altogether. This method, which we call ZETA (Zenith of Event-based Time-locked Anomalies), either outperformed or

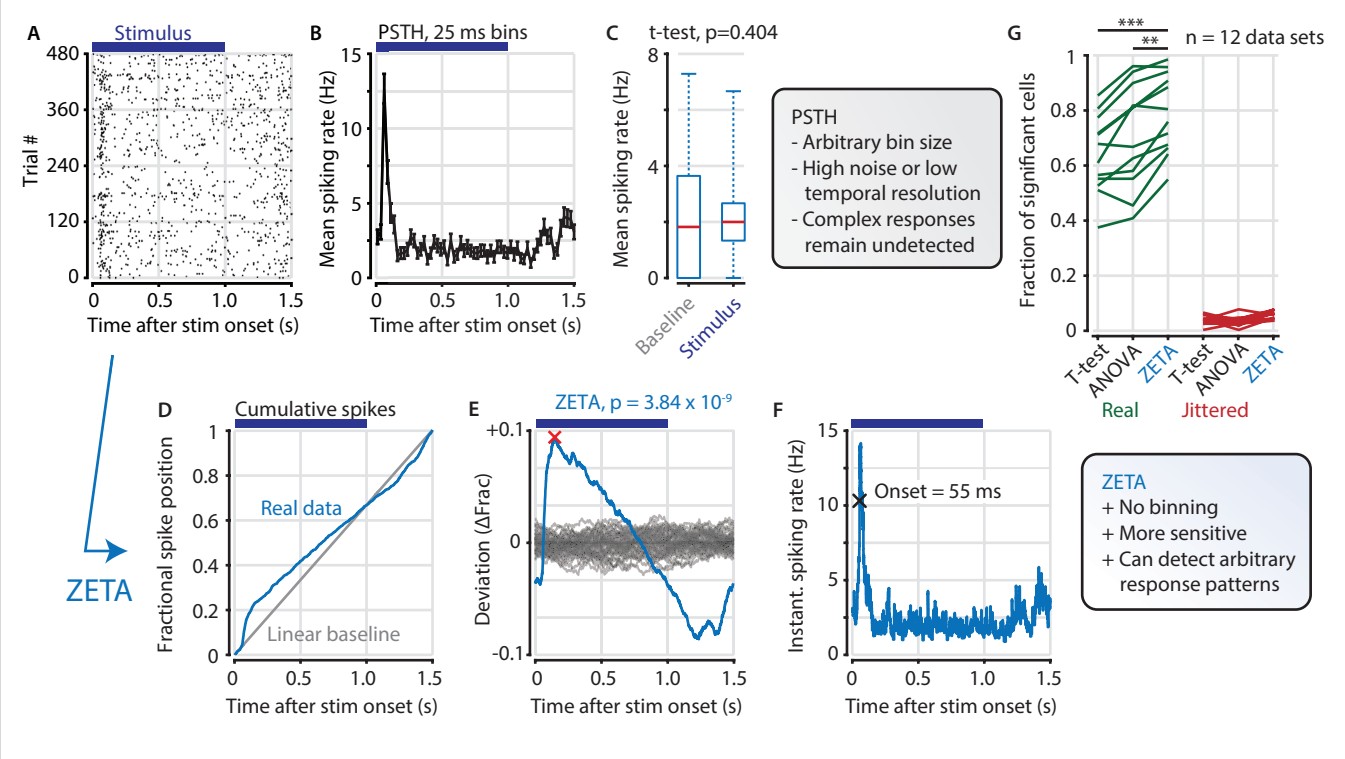

**Figure 1.** The ZETA-test detects an example visually responsive neuron in V1. (**A,B**) Raster plot (**A**) and PSTH (**B**) for a neuron that shows an onset peak and a reduced sustained spiking in response to a visual stimulus (purple bars). (**C**) A common approach for computing a neuron's responsiveness is to perform a t-test on the average activity per trial during stimulus presence (0–1 s) and absence (1.0–1.5 s), which fails to detect this neuron's response (paired t-test, p = 0.404, n = 480 trials). (**D**) ZETA avoids binning, by using the spike times to construct a fractional position for each spike (blue) and compares this with a null-distribution of a stationary rate (grey). (**E**) The difference between the real and null curves gives a deviation from expectation (blue), where the most extreme value is defined as the Zenith of Event-based Time-locked Anomalies (ZETA, red cross). To compute its statistical significance (here, p = 3.84 × 10⁻⁹), we compare ZETA to the variability over repeats of the procedure with jittered onsets (grey curves). (**F**) A ZETA-derived instantaneous spiking rate allows a reliable estimation of response onset. (**G**) At a significance threshold of α = 0.05, ZETA detects more stimulus-responsive cells than both a mean-rate t-test (***: paired t-test, p = 2.8 x 10⁻⁷, n = 12 data sets) and an ANOVA over bins using an optimal bin width (**: p = 0.0014).

matched that of t-tests, ANOVAs and point-process-based methods in all conditions tested. Building upon this framework, we also present a procedure to visualize instantaneous spiking rates without the need for binning, and show how this can be used to estimate peak-activity latencies with sub-millisecond accuracy. We apply these methods to transient-detected two-photon calcium imaging data from the visual cortex of mice traversing a virtual linear track and find that visuomotor mismatch signals and spatial location are encoded by different V1 neuronal subpopulations. Finally, we apply our approach to Allen Brain Institute Neuropixels data and show that optogenetic stimulation of VIP-expressing cells in mouse visual cortex has a separable early inhibitory and late disinhibitory effect on the local neural circuit. We anticipate that the ZETA-test will be a useful resource for a wide range of applications across various disciplines.

## Results
### ZETA: Zenith of event-based time-locked anomalies

A common procedure in pre-processing neural data is removing cells that are not responsive to an experimental stimulus. Many experimenters determine the 'stimulus responsiveness' of a cell by comparing its average spiking rate during the presentation and absence of a stimulus (see *Figure 1A–C* for an example V1 cell). This procedure will therefore remove neurons that show no response, but has the risk of also removing neurons that show a strong, but complex time-locked response to stimuli. To remedy this shortcoming, we developed a binning-free method for determining whether a neuron

shows any time-locked modulation. We call this statistical test ZETA for Zenith of Event-based Time-locked Anomalies (*Figure 1D–F*). It represents whether a neuron's spike train could be observed by chance, if it were not responding to an experimenter's event of interest: for example, the presentation of a visual stimulus, the onset of optogenetic stimulation, or a self-generated variable, such as an animal's location on a track.

ZETA is calculated on a single cell by performing the following steps. First, we align all spikes to stimulus onsets, as when making a raster plot (*Figure 1A*). Pooling all spikes across trials, we obtain a single vector of spike times relative to stimulus onset, and calculate the cumulative distribution as a function of time (*Figure 1D*). The deviation of this curve from a linear baseline represents whether the neuron has a higher or lower spiking density relative to a non-modulated spiking rate (*Figure 1E*, blue curve). We compare this pattern to the likelihood of observing it by chance by running multiple bootstraps by jittering stimulus-onset times to generate a null hypothesis distribution (*Figure 1E*, gray curves). After scaling the experimentally observed curve to the variation in the null hypothesis distribution, we use it to obtain a p-value corresponding to the Zenith of Event-based Time-locked Anomalies. Low ZETA-test p-values indicate that the neuron's firing pattern is statistically unlikely to be observed if the neuron is not modulated by the event of interest.

The ZETA-test bears similarities to a mean-subtracted Kolmogorov-Smirnov test applied to a renewal process model (see methods). In the method section 'ZETA and renewal-process models'

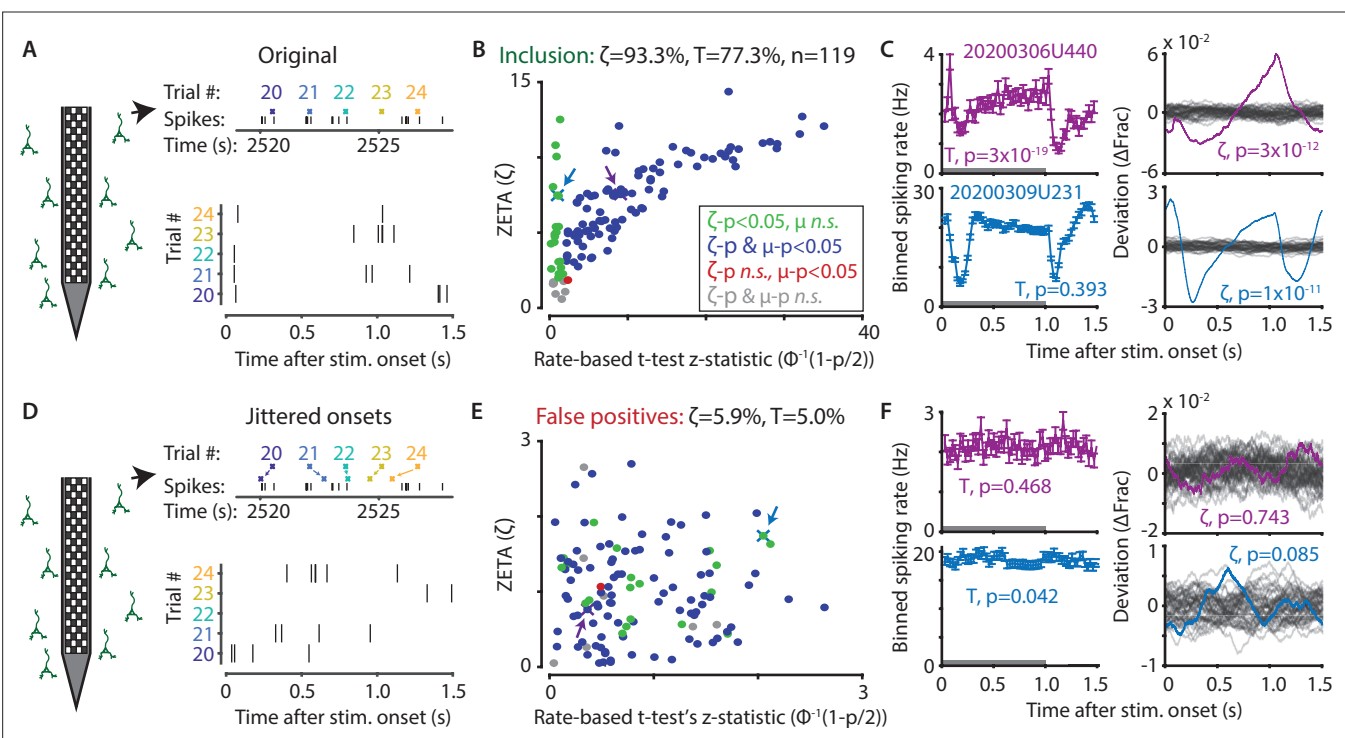

**Figure 2.** The ZETA-test outperforms the t-test for V1 neurons (n = 119). (**A**) Spikes were recorded with neuropixels in mouse V1 and aligned with the onset of square-wave drifting gratings. (**B**) Applying the ZETA-test, we found that 93.3 % of all V1 neurons showed significant firing rate modulations that were time-locked to stimulus onset. Using a rate-based t-test between stimulation (0–1 s) and no stimulation (1–1.5 s) epochs across trials registered 77.3 % of neurons to be visually responsive. Neurons detected by only ZETA are green, by both methods are blue, by only a mean-rate t-test are red, and by neither are grey. Arrows indicate the neurons shown in C. (**C**) Two example neurons that are (1) detected by a rate-based t-test as well as ZETA (top) and (2) not detected by a rate-based t-test, but only by ZETA (bottom). (**D**) We investigated the false-positive rate of both approaches by jittering the onsets of visual stimuli; this preserved the temporal structure of the spiking response, but destroys the time-locked modulations in activity. (**E–F**) same cells and analyses as in B-C but for jittered onsets. Red indicates neurons included by ZETA, but not a t-test; green indicates neurons included by a t-test, but not ZETA. As expected, the percentage of false positives (i.e., neurons with ζ /z-statistic > 2.0) was around 5 % (α = 0.05) for both approaches. Note the change in axis magnitude from B to E.

The online version of this article includes the following figure supplement(s) for figure 2:

**Figure supplement 1.** Several important variables underlying the ZETA-test can be derived analytically in the case of exponentially-distributed inter-spike intervals.

we show how the ZETA-test gains robustness to violations of the assumptions underlying renewal processes and outperforms alternative approaches. In short, the ZETA-test's main difference from other approaches used to infer a neuron's stimulus responsiveness, is that our test makes no a priori assumptions about the underlying distribution of temporal modulations and is binning-free. It can therefore detect both long-timescale changes in mean firing rate, as well as short-timescale stimulus-locked bursts or lapses of activity at any point in time relative to stimulus onset.

## Benchmarking the ZETA-test

To investigate whether the ZETA-test includes more cells recorded in mouse visual cortex in response to a drifting grating, while still retaining a 5 % false-positive rate at a significance level of $\alpha = 0.05$, we used a benchmarking test comparing onset-jittered and non-jittered data (*Figure 2*). In the non-jittered case, we compared the inclusion rate of the ZETA-test, as described above, to that of a mean-rate t-test. For the t-test, we calculated the average spiking rate of a cell during stimulation (0 s – 1 s after onset) and after stimulation (1 s – 1.5 s), and performed a paired t-test over trial repetitions (*Figure 1C*). This showed that cells included with a t-test were almost exclusively a subset of the cells detected with ZETA (see *Figure 2B* for all V1 cells recorded with Neuropixels). In other words, if a cell is detected as being visually responsive with a t-test, it is almost guaranteed to also be detected by the ZETA-test. In addition to these cells, the ZETA-test also includes cells that were not registered by a t-test. Although many varieties exist, *Figure 2C* shows an example cell that is detected by both t-tests and ZETA (top, sustained change in firing rate), and an example only detected by ZETA (bottom, balanced on/off peaks). In general, any cell lacking a sustained change but displaying a temporally non-uniform spiking distribution would be picked up by the ZETA-test but not a t-test: for example,

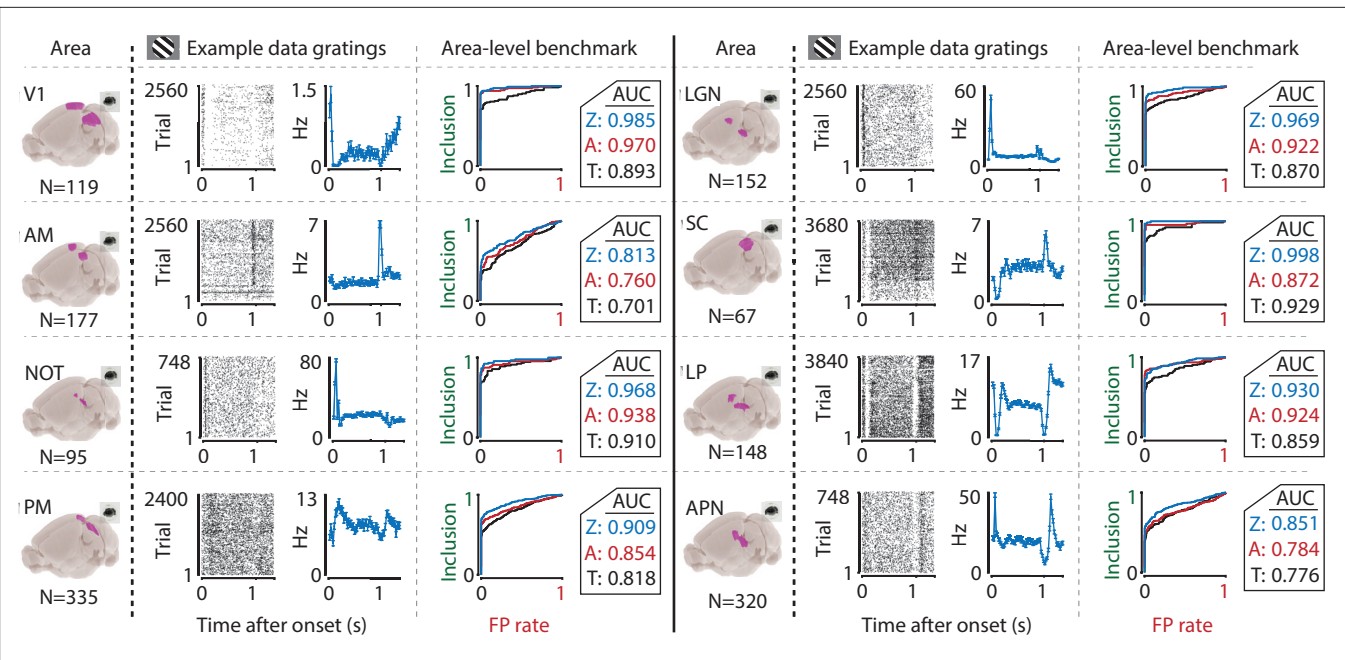

**Figure 3.** The ZETA-test's superior sensitivity is independent of brain area. We recorded neuronal responses to drifting gratings (1 s with 500 ms blank ITIs) from various visual brain areas: V1, SC, AM, LP, NOT, APN, PM, and LGN. For each area, we show an example neuron's raster plot (left) and binned responses (right). All cells depicted here were significant using ZETA. Area-level benchmark summaries show ROC analyses of the inclusion rate (y-axis) and false-positive (FP) rate (x-axis) for ZETA (Z, blue), bin-wise ANOVA (A, red), and rate-based t-tests (T, black). In all cases, the AUC of the ZETA-test exceeded both the ANOVA's AUC and t-test's AUC.

The online version of this article includes the following figure supplement(s) for figure 3:

**Figure supplement 1.** The ZETA-test is more sensitive than a mean-rate approach across a range of areas, recording techniques and stimuli.

**Figure supplement 2.** Model-based methods require high spike counts or hyperparameter tuning.

**Figure supplement 3.** Intermediate statistical tests reveal the functions of the components of the ZETA-test.

**Figure supplement 4.** The ZETA-test performs better than t-tests at low trial numbers and regardless of data preparation.

cells with a sharp but narrow onset peak and variable baseline activity, cells with a balanced on/off response, or oscillatory cells that phase-reset on stimulus onset.

Next, we ran the same tests again, but now on data where we randomly jittered the stimulus onset times between $-\tau$ and $+\tau$, where $\tau$ is median onset-to-onset duration. This procedure preserves the properties of a neuron's spike train, but removes locking of the responses to the stimulus (*Figure 2D*). If the ZETA-test simply always gave low p-values, then this would result in a high false-positive rate as many neurons would still be included. In contrast, the false-positive rate of the ZETA-test was generally low, and consistent with that expected for a significance level of $\alpha = 0.05$ (*Figure 2E–F*).

## Sensitivity of ZETA is superior to mean-rate T-tests

We performed this benchmark for single-cell activity obtained from n = 12 combinations of various visual regions (V1, AM, PM, LGN, SC, LP, NOT, APN, Retina) using multiple techniques (Neuropixels, n = 8; Neuronexus, n = 2; GCaMP6, n = 1; pMEA, n = 1), in response to light flashes (retina, n = 1) or drifting gratings (all others, n = 11). Under all conditions, the inclusion rate using ZETA-tests was higher than using t-tests: at a significance level of $\alpha = 0.05$, the inclusion rate for the ZETA-tests was 79 % and for mean-rate t-tests was 64 %; t-test of ZETA vs mean-rate t-test inclusion rates: n = 12 data sets, p = $2.8 \times 10^{-7}$ (*Figures 1G and 3*, *Figure 3—figure supplement 1*). This means that the ZETA-test includes 42 % of the cells that were not included by a t-test. A significance level of $\alpha = 0.05$ is rather arbitrary, so we also performed a receiver operating characteristic (ROC) analysis, where we investigated the number of inclusions as a function of the number of false positives (*Figure 3*). The ROC's summary statistic is the area under the curve (AUC); one being a perfect discriminator. Again, we found that the ZETA-test showed a higher statistical sensitivity (ZETA-test, mean AUC = 0.914) than a mean-rate t-test (t-test, mean AUC = 0.843), and that this difference was statistically significant (paired t-test, n = 12, p = $4.9 \times 10^{-6}$).

We also benchmarked various versions of tests derived from the theoretical framework of renewal and Poisson process models. None of these models reached the statistical power and computational efficiency of the ZETA-test (*Figure 3—figure supplements 2 and 3*), but they do provide an attractive mathematical connection to a more widely studied class of models. We have therefore provided more information on the mathematical relationship between this class of models and the ZETA-test in the method section.

The percentage of visually responsive cells detected by the ZETA-test is higher than typically reported. For example, using gratings that only differed by orientation, we found 93.3 % of all V1 cells to be visually-modulated. Even more striking was that the lower bound of the binomial 95%-confidence interval was at 89.7 %. This lower-bound is higher than the responsiveness previously reported in many studies, including our own (*Montijn et al., 2016a*; *Niell and Stryker, 2008*; *Shuler and Bear, 2006*; *Steinmetz et al., 2019*), exemplifying the advantage of the ZETA-test.

The data used so far for benchmarking contains more stimulus repetitions than commonly used in neuroscience. This raises the possibility that the ZETA-test is only advantageous when large numbers of trials are used. To test this, we randomly subsampled the number of trials included in the analysis, and repeated our benchmark. As the results for V1 in *Figure 3—figure supplement 4A* show, the ZETA-test consistently included more than the t-test, regardless of the number of trials.

We hypothesized that the t-test's worse performance might result from pooling the responses to different orientations into one group. Therefore, we repeated the t-test's benchmark after first splitting the trials into 24 groups corresponding to the 24 directions we presented. A neuron was included if the spiking rate during stimulus presentation was significantly different from its pre-stimulus baseline rate in any group, after applying a Bonferroni correction. However, this procedure reduced the t-tests' performance (*Figure 3—figure supplement 4A*, middle panel).

We noticed that the t-test's false positive rate was rather low after Bonferroni corrections. To test whether we over-corrected the t-test, we removed the multiple-comparison correction (*Figure 3—figure supplement 4A*, right-hand panel). In this case, the t-test false positive rate increased to >50%, while its inclusion rate (89.9%) remained lower than that of the ZETA-test (95.0%). Finally, we investigated whether the t-test's performance was hampered by including the immediate off-response after stimulus offset in the baseline period. We reran the above analyses, but now limited the baseline to the 300 ms preceding the stimulus onset. As can be seen in *Figure 3—figure supplement 4B*, this did not improve the t-test's performance. In summary, the t-test is at its most sensitive when using

the full 500 ms epoch in-between stimulus presentations as baseline period and when pooling data across all orientations.

## The sensitivity of ZETA is superior to an ANOVA with an optimal bin width

Mean-rate t-tests are common in neuroscientific analysis, but it could be argued that this is somewhat of a strawman to use as baseline performance. An alternative is to construct a peri-stimulus time histogram (PSTH) and run a one-way ANOVA across bins to test a neuron's responsiveness to a particular stimulus. However, because this requires picking a bin width, this can lead to arbitrary choices based on the experimenter's visual inspection of the data, which might increase false positive rates. A better solution is to use one of the various methods to estimate optimal widths for binning (*Freedman and Diaconis, 1981*; *Scott, 2009*; *Shimazaki and Shinomoto, 2007*). We therefore calculated the optimal bin width using the Shimazaki & Shinomoto method, which was specifically designed for building a PSTH, and repeated the benchmark described above, but now testing the responsiveness of neurons using an ANOVA (see Materials and methods). This ANOVA procedure performed markedly better than a mean-rate t-test (*Figures 1G and 3*). However, we found that it still showed an inclusion rate (at $\alpha$ = 0.05) that was lower than using the ZETA-test (ANOVA mean inclusion rate = 71 %; ZETA-test inclusion = 79%, paired t-test, n = 12, p = 0.0014). Importantly, this difference could not be explained by different levels of false positives, as an ROC analysis also showed a superior statistical sensitivity for the ZETA-test: mean ANOVA-AUC = 0.880, mean ZETA-AUC = 0.914, paired t-test, n = 12, p = $7.7 \times 10^{-4}$.

Taken together, the results of comparing the ZETA-test to t-tests, ANOVAs, and renewal-process based tests show that the binless ZETA-test has a statistical sensitivity superior to all alternative tests, regardless of number of trial repetitions, brain region where the data were recorded, or specifics of the data preparation.

## ZETA-test in the absence of short peaks of activity

Having established that the ZETA-test performs well in real neural data, we looked for conditions under which the ZETA-test fails. We know that the t-test has access to information that the ZETA-test does not: the spike times used by the ZETA-test are flattened over trials, while the t-test uses the variability across trials. Therefore, when the variability of mean activity across trials is low, but the variability of spike times within a trial is high, the ZETA-test could perform worse than a t-test.

To test this hypothesis, we simulated Poisson-spiking artificial neurons (see Materials and methods) where we changed two variables: (1) we varied the difference in spiking rate (dHz) between stimulus and baseline (both 1 s) from 0 to 1 Hz with a background rate of 1 Hz and (2) we varied the period over which the neuron was active during stimulus presentation ($T_r$) from 0.5 to 1.0 s while keeping the total spike count constant. In effect, neurons with $T_r$ <1.0 show a tri-phasic response, first increasing their spiking rate, then showing a cessation of activity, and finally returning to baseline. As before, we compared the tests' performance using an ROC analysis. The ROC's area under the curve (AUC) is conveniently similar to a Wilcoxon-Mann-Whitney statistic and can be used to directly determine which of two procedures is more sensitive (*Calders and Jaroszewicz, 2007*). As expected (*Figure 4A*), the t-test's AUC depends only on dHz and fails when dHz = 0, while the ZETA-test also discriminates well when there is no difference in spike counts, but there is a consistent temporal discontinuity (dHz = 0, $T_r$ <1.0). Interestingly, while the t-test performs better than the ZETA-test when $T_r$ = 1.0, the ZETA-test also still performs reasonably well (*Figure 4B*).

While this scenario is important to consider from a theoretical perspective, pure Poisson-spiking neurons probably do not exist in the brain. We therefore proceeded with a (somewhat) more biologically plausible simulation of bursting cells, where their bursting probability is orientation-tuned (*Figure 4C*, see Materials ad methods and *Table 1*). These neurons show no consistent peaks or troughs of activity (*Figure 4D and E*). However, the highly variable spike counts this bursting produces result in the ZETA-test outperforming the t-test (AUC, ZETA = 0.941, t-test = 0.902, z-test, p = $4.1 \times 10^{-11}$). To conclude, even in hypothetical scenarios that we specifically constructed to investigate the limits of the ZETA-test, it performs close to the t-test (*Figure 4B4*). Importantly, in the case of strongly bursting cells (*Figure 4F*), the ZETA-test clearly outperforms the t-test.

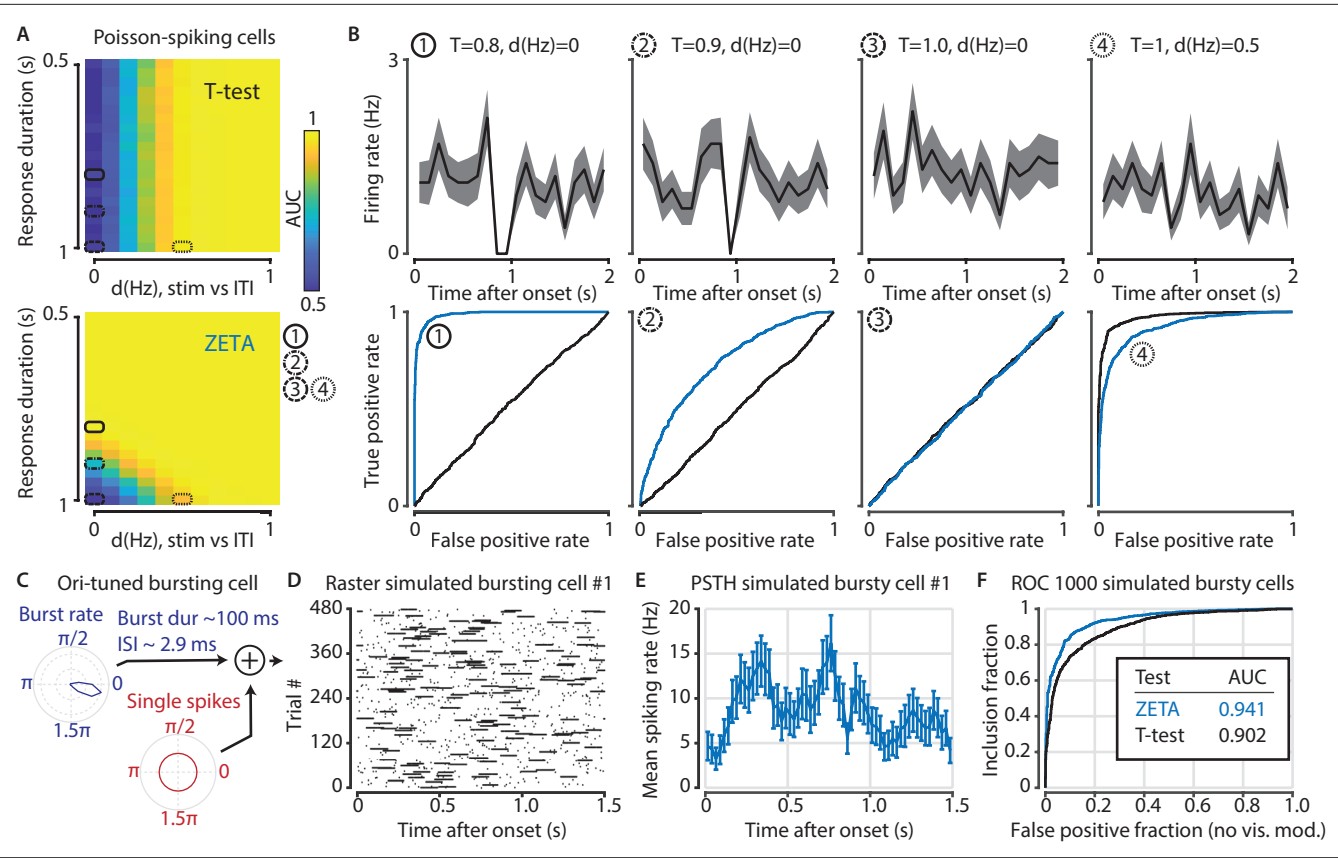

**Figure 4.** The t-test exceeds the ZETA-test's performance only in the hypothetical case of purely Poisson-distributed spike counts and only for a small range of spiking rate differences. (**A**) Top: the ability of the t-test to differentiate simulated stimulus (non)modulated cells depends only the total spike count during stimulus (0–1 s) and inter-trial interval (ITI, 1–2 s) periods (x axis; firing rate difference *d(Hz)*), but not on the duration of the cell's response when keeping the spike count constant (y axis; response duration *T*). Bottom: in contrast, the ZETA-test can differentiate stimulus modulation using either variable. (**B**) Top: example PSTHs of single cells corresponding to the markers 1–4 in (**A**). Bottom: ROC curves for the combination of variables marked as 1–4 in (**A**). The ZETA-test always exceeds the t-test's performance, except for a limited range where the response duration is 1 s. (**C**) A more biologically plausible test case is bursting neurons that have an elevated probability of bursting during stimuli. (**D–E**) This simulation produces no 'onset' peaks of activity that the ZETA-test can exploit. (**F**) However, despite the lack of clear peaks of activity, the ZETA-test exceeds the t-test's ability to detect stimulus-modulated bursting cells (ZETA AUC = 0.941, t-test AUC = 0.902).

**Table 1.** Parameters of bursting neurons used in *Figure 4*.
Abbreviations and mathematical symbols are as follows: ISI = Inter-spike interval; IBI = Inter-burst interval; Exp = exponential distribution; |x| = absolute of x; N = standard normal distribution; U(x,y) = uniform distribution on interval [x,y]; $\mathcal{M}$ = von Mises distribution; $\Gamma$ = Gamma distribution.

| Property | Unit | Distributed as | Sampled from: |
|---|---|---|---|
| Single-spike ISI | s | $Exp(1/r)$ | $r \sim Exp(\lambda = 1)$ |
| Baseline IBI | s | $Exp(1/|R_b|)$ | $R_b \sim |N|/20 + 1/80$ |
| Preferred orientation IBI | s | $Exp(1/|R_t|)$ | $R_t \sim |N| + 1/4$ |
| Preferred orientation | rad | $\theta_p$ | $\theta_p \sim U(0,2\pi)$ |
| Orientation-tuned bursting | Hz | $1/R_b + 1/R_t \cdot \mathcal{M}(\theta_p, \kappa)$ | $\kappa \sim 5 + U(0,5)$ |
| Burst duration | ms | $\Gamma(2*k, \theta = 0.5)$ | $k \sim 90 + 10*N$ |
| ISI in bursts | ms | $\Gamma(2*k, \theta = 0.5)$ | $k \sim 0.5 + Exp(\lambda = 2.4)$ |

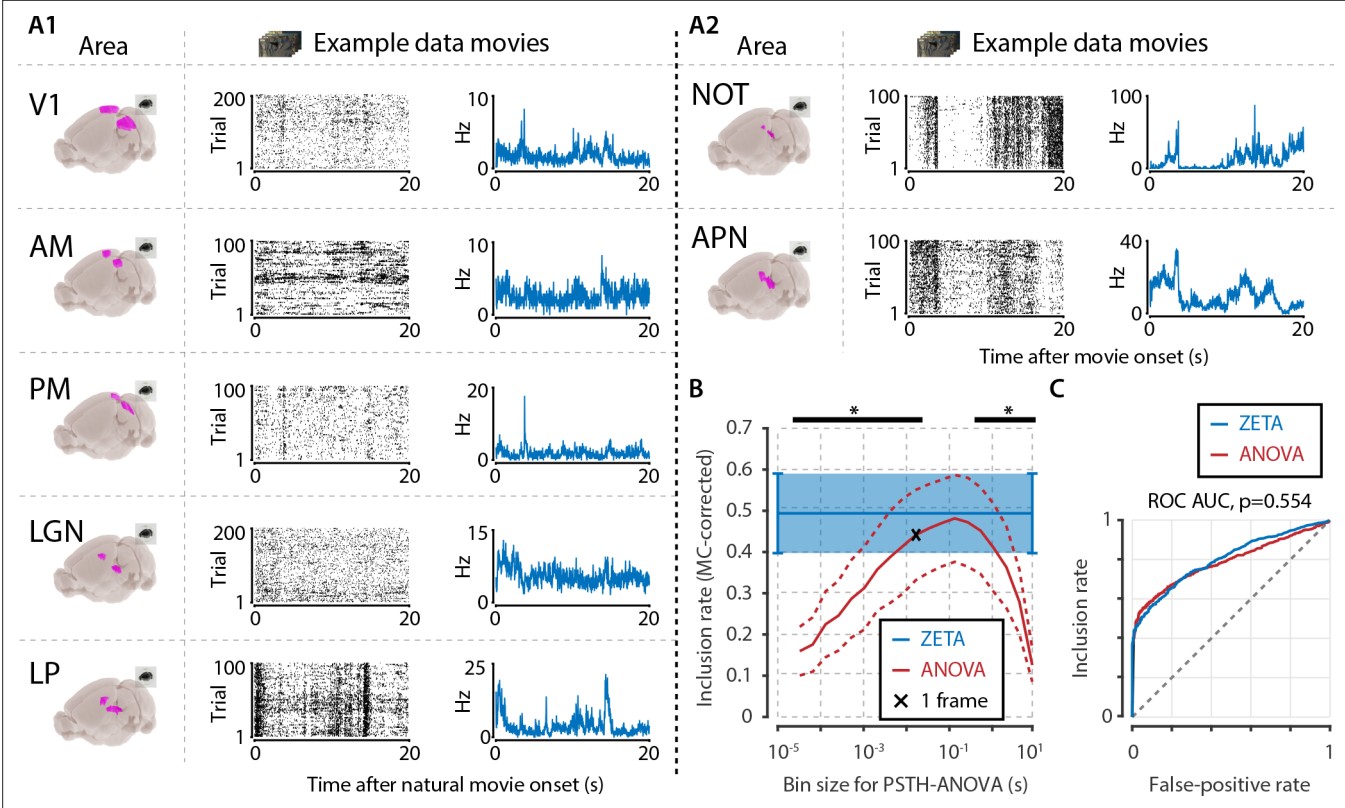

**Figure 5.** Determining neuronal responsiveness to natural movies. (**A**) We recorded neuronal responses to natural movies (four scenes that repeated every 20 s) from various visual brain areas: V1, AM, PM, LGN, LP, NOT, and APN. For each area, we show an example neuron's raster plot (left) and binned responses (right). All cells depicted here were significant using ZETA. (**B**) We determined a neuron's responsiveness using a 1-way ANOVA over all binned responses (i.e. PSTH-level ANOVA) for various bin sizes and MC-corrected (black) and using the ZETA-test (blue). Note that the ZETA-test is timescale-free and plotted at all x-values only for easy comparison with the ANOVAs. Curves show mean ± SEM over brain regions (n = 7). ZETA shows an inclusion rate similar to the most optimal bin sizes (0.0333s – 0.2667s), and significantly higher than bin sizes of 0.0167 s or shorter, as well as 0.533 s or longer (*, FDR-corrected paired t-tests, p < 0.05). (**C**) An ROC analysis on all n = 977 cells showed that the ZETA-test and a combined set of ANOVAs were similarly sensitive (ZETA-test AUC: 0.792 ± 0.011; ANOVAs AUC: 0.798 ± 0.010; z-test, p = 0.554).

## Neuronal responsiveness to natural movies

Next, we asked how the performance of the ZETA-test compares to that of an ANOVA, in a case where there is no a priori knowledge regarding the neuronal response profile, but where the stimulus itself provides a natural timescale that may be used for binning neuronal responses. We therefore determined the responsiveness of neurons to natural movies, using either the ZETA-test, or a one-way ANOVA across bins, repeated for different bin sizes (i.e. timescales). Single-cell data were recorded using Neuropixels in seven visual brain areas of 3 mice, while the animals were presented with repetitions of 20 s long natural movies (*Figure 5A*).

To ensure that the ANOVA approach could detect short bursts of activity, as well as long timescale whole-scene modulations of firing rates, we chose a wide range of bin sizes. We picked a single movie frame duration (0.0167 s) as the centre point, and used bins from 1/512th up to 512 movie frames, spaced equidistantly on a base-2 logarithmic scale in 18 steps. For each area, we pooled all cells, and calculated the area-level inclusion rate with either the timescale-free ZETA-test or Bonferroni-corrected ANOVAs at different timescales (*Figure 5B*). Bin sizes of 33–267 ms did not differ significantly from the ZETA-test's inclusion rate (FDR-corrected paired t-tests, p > 0.05, n = 7 areas), while the ZETA-test's inclusion rate was higher than with an ANOVA for all short ( < 33 ms) and long ( > 267 ms) bin durations (p < 0.05).

The above approaches give some insight into which bin sizes best capture the dominant temporal components in neuronal responses in our data, but a more powerful approach might be to classify a neuron as "included" whenever any of the 19 ANOVAs reached significance (i.e. p < α). Repeating this

procedure for various significance levels α on the interval (0,1) produces an ROC curve (*Figure 5C*). Using this approach, we found there was no significant difference in performance between a set of ANOVAs and the ZETA-test (z-test, p = 0.554): ANOVAs AUC = 0.798 ± 0.010, ZETA-test AUC = 0.792 ± 0.011 (mean ± sd). Overall, the above results show that, under these conditions, the binless and timescale-free ZETA-test performs as well as an aggregate set of ANOVAs binned at various timescales.

## Instantaneous firing rates (IFRs) for visualization and onset latency detection

The above paragraphs have shown that the ZETA-test is a sensitive statistical tool to detect whether neurons respond to a stimulus. However, it cannot be used to determine when exactly the strongest

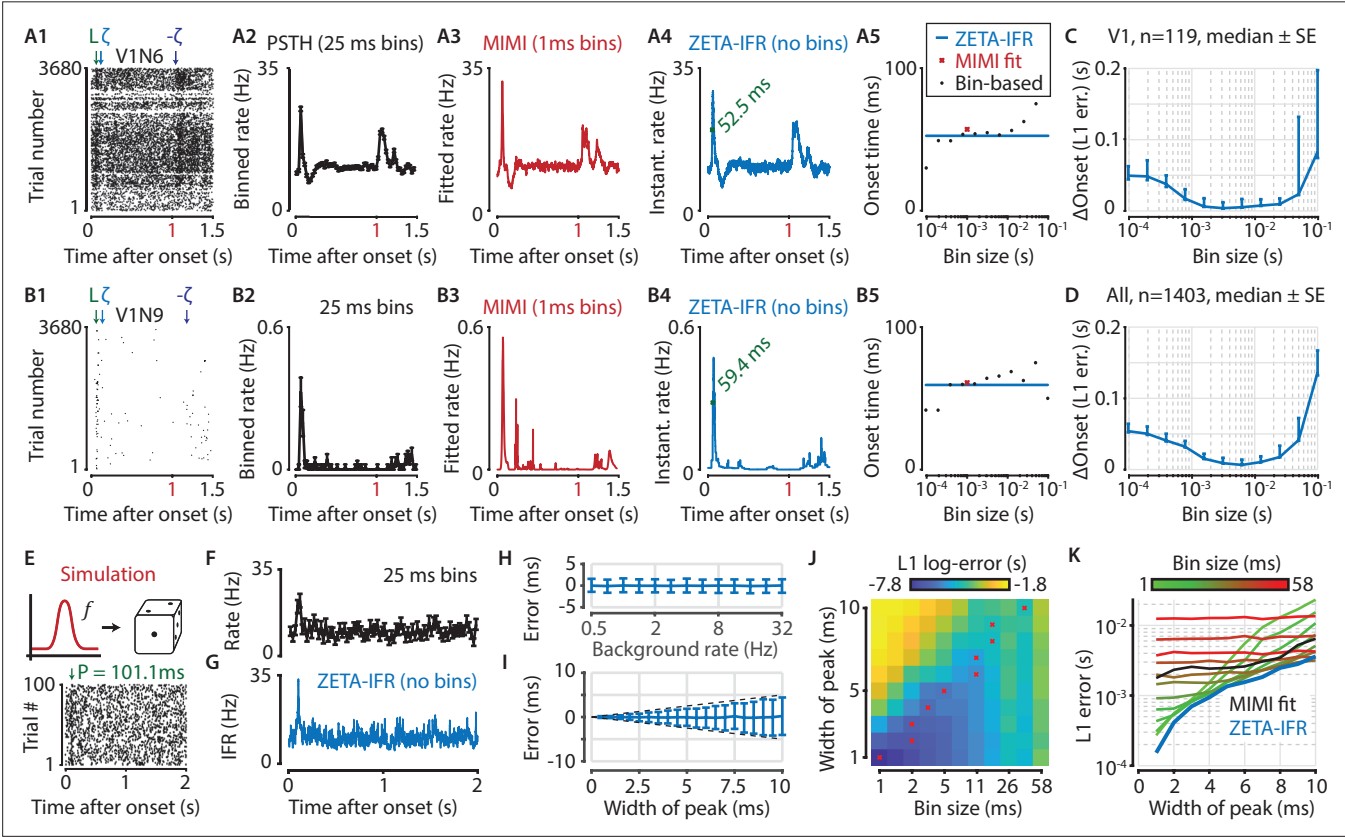

**Figure 6.** A ZETA-derived measure for instantaneous firing rates (IFR) avoids binning and allows more accurate latency determination than with PSTHs. (**A,B**) Responses of two example V1 cells to drifting gratings. From left to right: (1) Raster plots showing the estimated onset latency ('L', green), and times of ZETA (blue) and –ZETA (purple). (2) Spiking rates using 25 ms bins. (3) Spiking rates using multiplicative inhomogeneous Markov interval (MIMI) model-based fits. (4) Binning-free instantaneous spiking rates provide a much higher temporal resolution. Using the first crossing of half-peak firing rates, we determined the onset latencies of these cells to be 52.5 and 59.4 ms. (5) Estimated onset times using our method for instantaneous firing rates (blue), MIMI model-based fits (red), or PSTHs with bin widths from 0.1 to 100 ms (black). Onset latency estimates depend on the chosen bin size, and the optimal size varies across cells. (**C–D**) The median (± SE) difference in onsets estimated by our instantaneous firing rates compared to that of various sized bins for V1 cells (C, n = 119) and for cells from all brain regions (D, n = 1403). Both C and D show the onsets estimated by the two methods were most similar for bin sizes between 1–10 ms. (**E–K**) Benchmarking of peak detection using artificial Poisson neurons that show a transient peak. (**E**) Example Poisson cell for a background rate of 10 Hz and a peak-width of 10 ms. With the true peak at 100 ms, the estimation error here was 1.1 ms. (**F**) Binning the cell's spiking response in 25 ms bins reduces the peak height and temporal precision. (**G**) Our instantaneous spiking rate preserves a sharper peak response and allows for a temporally accurate latency estimation. (**H**) The detection of peak latencies is insensitive to realistic levels of a stationary Poisson background firing rate (13 base rates, 0.5–32 Hz). (**I**) The mean error is unbiased, and the standard deviation in the onset peak latency estimate scales linearly with the width of the peak. Dotted lines show the real peak width. Graphs in H-I show mean ± sd. (**J**) The error in peak latency estimation depends on both the bin width and the width of the neuron's peak response. Red crosses indicate the bin size with the lowest error for a given peak width. (**K**) Plotting the latency estimation error shows that different bin sizes (red-green) are optimal for different peak-widths. The accuracy of the latency obtained from MIMI-based fits (black) is less sensitive to the peak width, but never performs as well as well as the most optimal bin size. The error based on our binning-less IFR (blue) is as at least as low as the most optimal bin size, for any peak width.

response of a neuron occurs. Therefore, we also developed a method that determines the instantaneous firing rate (IFR) using the temporal deviations upon which ZETA is based. Like the ZETA-test, it avoids the bin size selection issue of peri-stimulus time histograms (PSTHs). Another advantage of this IFR is that its temporal resolution is limited only by the neuron's spike density. Moreover, unlike model-based methods, such as the multiplicative inhomogeneous Markov interval (MIMI) model (*Kass and Ventura, 2001*), it requires no fitting and is orders of magnitude faster (see Materials and methods, *Figure 3—figure supplement 2*). It is therefore a useful tool for determining spike train features with high precision, such as a neuron's onset latency.

*Figure 6A and B* show two example V1 neurons with a relatively high (*Figure 6A*) and low (*Figure 6B*) firing rate. Here we define 'onset' as the time the half-maximal response of the peak is first crossed, a metric that is heavily influenced by the chosen bin size when using PSTH-based analyses. Moreover, this bin-width-dependent estimation makes PSTH-based comparisons across cell classes problematic, as the choice of optimal bin width depends on spiking properties such as the firing rate and peak firing duration (*Freedman and Diaconis, 1981*; *Scott, 2009*; *Shimazaki and Shinomoto, 2007*), which are heterogeneous across neuronal cell types (*Figure 6A5 and B5*). Hence, the main advantage of our method is that it avoids having to tailor the bin size to each neuron individually; allowing for better comparisons across varying cell types and brain regions.

To test performance on real data, we compared the estimated onset latency using our IFR to what we obtain from a PSTH analysis with varying bin sizes. Across all V1 neurons (*Figure 6C*, n = 119 neurons) as well as all neurons recorded in visual areas (*Figure 6D*, n = 1403 neurons), our metric showed the strongest agreement in latency estimation with PSTH bin sizes between 1 and 10 ms.

As no ground truth is known for the real latencies of experimentally recorded neurons, we performed a benchmark test with artificially generated spike trains. Like before, we used Poisson neurons with a constant background spiking rate, where each neuron was assigned a background rate ranging from 0.5 to 32 Hz (*Figure 6E–G*). We used 100 trials, each 2 s long, and created a peak response on top of this baseline rate by adding a single spike in 50 % of all trials (i.e. 50 spikes in total). We also varied the peak-response width by jittering the time each spike was added according to a normal distribution, σ ranging from 1 to 10 ms. We found that the error in the peak estimate was independent of background spiking rate (*Figure 6H*), indicating robust performance even when the spikes contributing to the peak were only 0.78 % of the total (i.e. for 32 Hz). The mean error was also independent of the jitter, while the standard deviation of the error estimate grew in a theoretically optimal fashion as $O(\sigma)$ (*Figure 6I*).

We next tested the estimator accuracy when using a binning-based PSTH approach (bin width 1–58 ms) and found it depends on both the bin width and the width of the peak response (*Figure 6J*). This means that accurate latency determination using PSTHs requires the use of multiple bin sizes. More importantly, when comparing the estimation error between these bin-based methods and our IFR, we found that the binning-less IFR-based latencies were consistently as accurate as, or more accurate than, the best possible bin-width for any given peak-width (*Figure 6K*). Finally, the IFR-latency accuracy also exceeded the MIMI-model fit based method. In other words, the IFR-based accuracy supersedes the PSTH-based (and MIMI-based) accuracy without the need to hand-pick the optimal binning width per neuron for a PSTH-method, nor tune hyperparameters such as knot number, location, and regularization strength for the MIMI-method.

## Visuomotor mismatch and spatial location are mediated by different neuronal subpopulations

Having established that the ZETA-test and IFR are statistically robust and have clear advantages over mean-rate approaches and PSTHs, we applied these tools to a GCaMP6 data set. Many theories, such as predictive coding (*Friston, 2005*; *Gregory et al., 1980*; *Rao and Ballard, 1999*), biologically realistic error backpropagation (*Ooyen and Roelfsema, 2003*; *Whittington and Bogacz, 2019*), and canonical cortical microcircuit operation (*Bastos et al., 2012*; *Douglas et al., 1989*), define a 'top-down' signal representing an expectation, error or surprise signal as distinct from a bottom-up sensory drive. In mouse V1, such top-down visuomotor mismatch signals have been reported previously (*Attinger et al., 2017*; *Keller et al., 2012*; *Leinweber et al., 2017*; *Saleem et al., 2013*). However, whether individual V1 neurons can be classified into different groups based on their encoding of top-down visuomotor mismatch or bottom-up sensory-driven spatial location signals has not been studied.

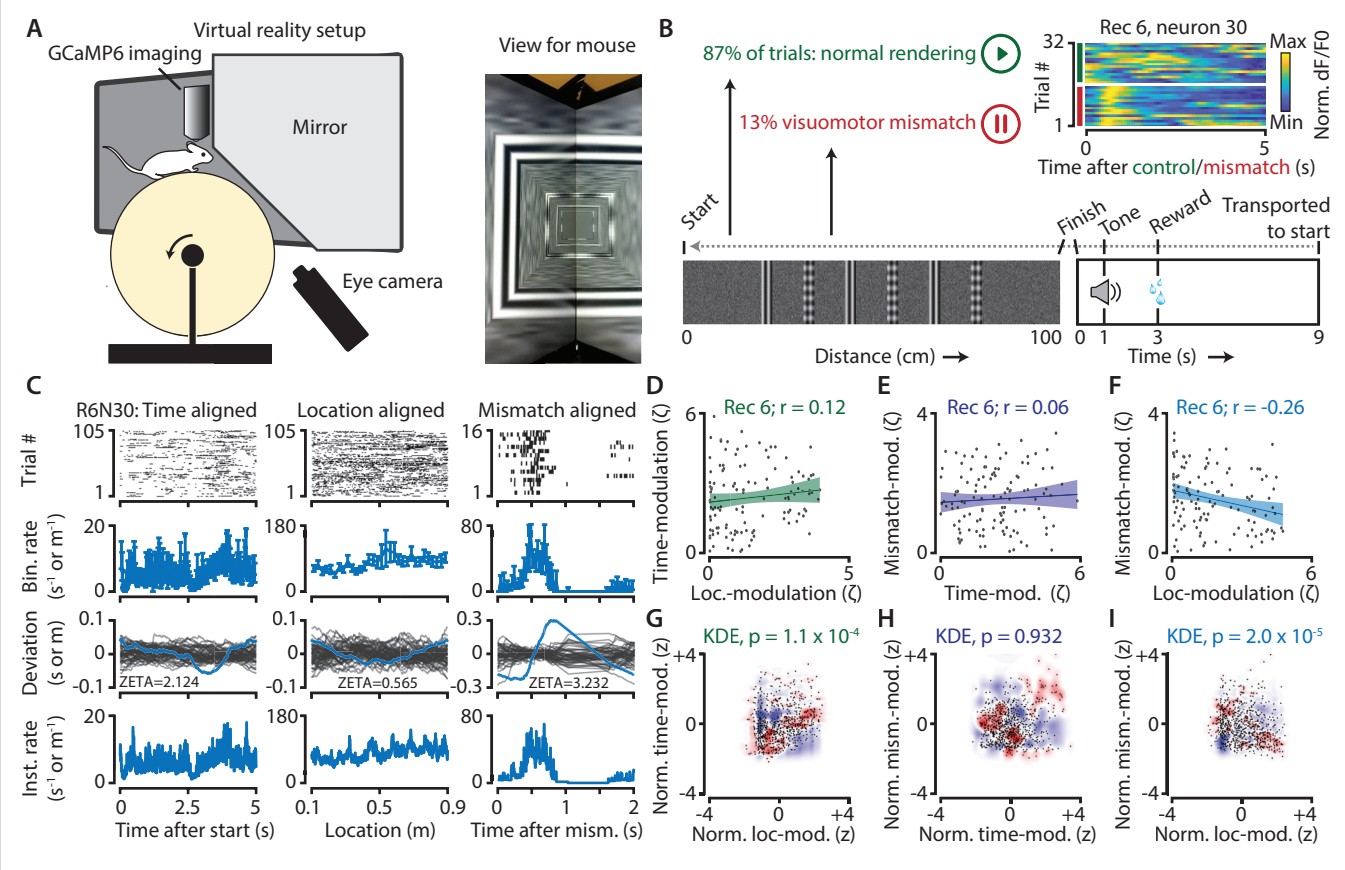

**Figure 7.** Neurons in V1 encode either visuomotor mismatch signals or spatial location. (**A**) Schematic of setup showing mouse on running wheel (lhs) viewing a virtual tunnel (rhs). (**B**) Trials consist of a 100 cm linear track. One second after the mice ran to the end of the tunnel, an auditory stimulus signaled that a water reward would be delivered two seconds later. 6 s after reward delivery, mice were transported to the start of the virtual tunnel. In a subset of trials, the rendering of the tunnel was paused at a random location, eliciting a visuomotor mismatch signal. Top right shows calcium imaging data for an example 'mismatch neuron' during 16 control and 16 mismatch trials. (**C**) Spiking data for example neuron obtained from exponential fits of the dF/F0 signals. Putative spikes were aligned to start (left), location of the animal on the track (middle), or mismatch onset (right). From top to bottom: raster plot of putative spike times; mean ± SEM of firing rates over trials (n = 105 trials, of which n = 16 mismatch trials); spiking deviation underlying ZETA; instantaneous firing rate. (**D–I**) Relationship between time-, location-, and mismatch-modulation. One point is one neuron. (**D–F**) ZETA-scores for example recording 6 (N = 120 neurons). (**G–I**) Analysis using a kernel-density estimate (KDE) to test whether joint-encoding of two features is more common than expected by chance (see **Figure 1**). (**G**) More neurons showed joint-encoding of both spatial and temporal location than expected by chance (p = 1.1 × 10⁻⁴). (**H**) Joint-encoding of temporal location and mismatch was not significantly different from chance (p = 0.932). (**I**) Location on the virtual track and visuomotor mismatch are less likely to be encoded by the same neuron than expected from chance (p = 2.0 × 10⁻⁵). See **Figure 7— figure supplement 1** for more details on the KDE procedure.

The online version of this article includes the following figure supplement(s) for figure 7:

**Figure supplement 1.** Using a kernel-density estimator (KDE) to investigate the joint-encoding of two features.

To examine this issue, and to provide an example of how one could use ZETA in a neurophysiology study, we used neuronal calcium data recorded in L2/3 V1 of 4 mice running on a virtual-reality linear track (**Figure 7A**). In 87 % of all corridor runs (N = 622/713 trials), the track was rendered normally, and the mice received visual feedback matching their running speed. In the remaining 13 % of runs (N = 91), rendering was halted at a random location for 500 ms before resuming (**Figure 7B**). After performing calcium-transient detection to obtain putative spike times (**Montijn et al., 2016b**), we calculated ZETA-scores for all neurons in three different ways. We aligned the spikes to mismatch-onsets, to trial starts, or converted the spike times into locations on the track, and aligned these spike locations to the start. For each recording (n = 7), we calculated the Pearson correlation for each pair of these three ZETA-scores (**Figure 7C–F**). Across recordings, time- and location-modulation were positively correlated (mean r = 0.23, one-sample t-test, n = 7 recordings, p = 0.016); time- and

mismatch-modulation were not significant ($r = 0.13$, $p = 0.28$); and location- and mismatch-modulation were negatively correlated ($r = −0.22$, $p = 0.04$).

We next used a kernel-density estimate (KDE) to directly test whether the joint encoding of two features within single neurons was different from chance (*Figure 7G–I*; *Figure 7—figure supplement 1*). Indeed, we found that it was less likely that a neuron showed high modulation values for both spatial location and visuomotor mismatch than if the two features were encoded independently ($z = −4.3$, $p = 2.0 \times 10^{-5}$). We also found time and location to be more likely to be encoded by the same neurons ($z = 3.8$, $p = 1.1 \times 10^{-4}$), while time- and mismatch-modulation show no effect ($z = −0.06$, $p = 0.95$). This suggests a functional specialization for many neurons to encode either visuomotor mismatch signals or spatial location, but not both. While it is possible that location- and mismatch-encoding are also mediated by specific genetic subtypes of interneurons (*Attinger et al., 2017*), our analysis of putative pyramidal cells demonstrates that principal cells also show encoding specialization.

### Optogenetic stimulation of VIP cells disinhibits visual cortex

Finally, we applied the IFR and ZETA-test to data recorded at the Allen Brain Institute (*Siegle et al., 2019*). In this case, we investigate whether optogenetic stimulation of VIP cells in visual cortex disinhibits the local circuit, as has been shown previously for auditory cortex and mPFC (*Pi et al., 2013*).

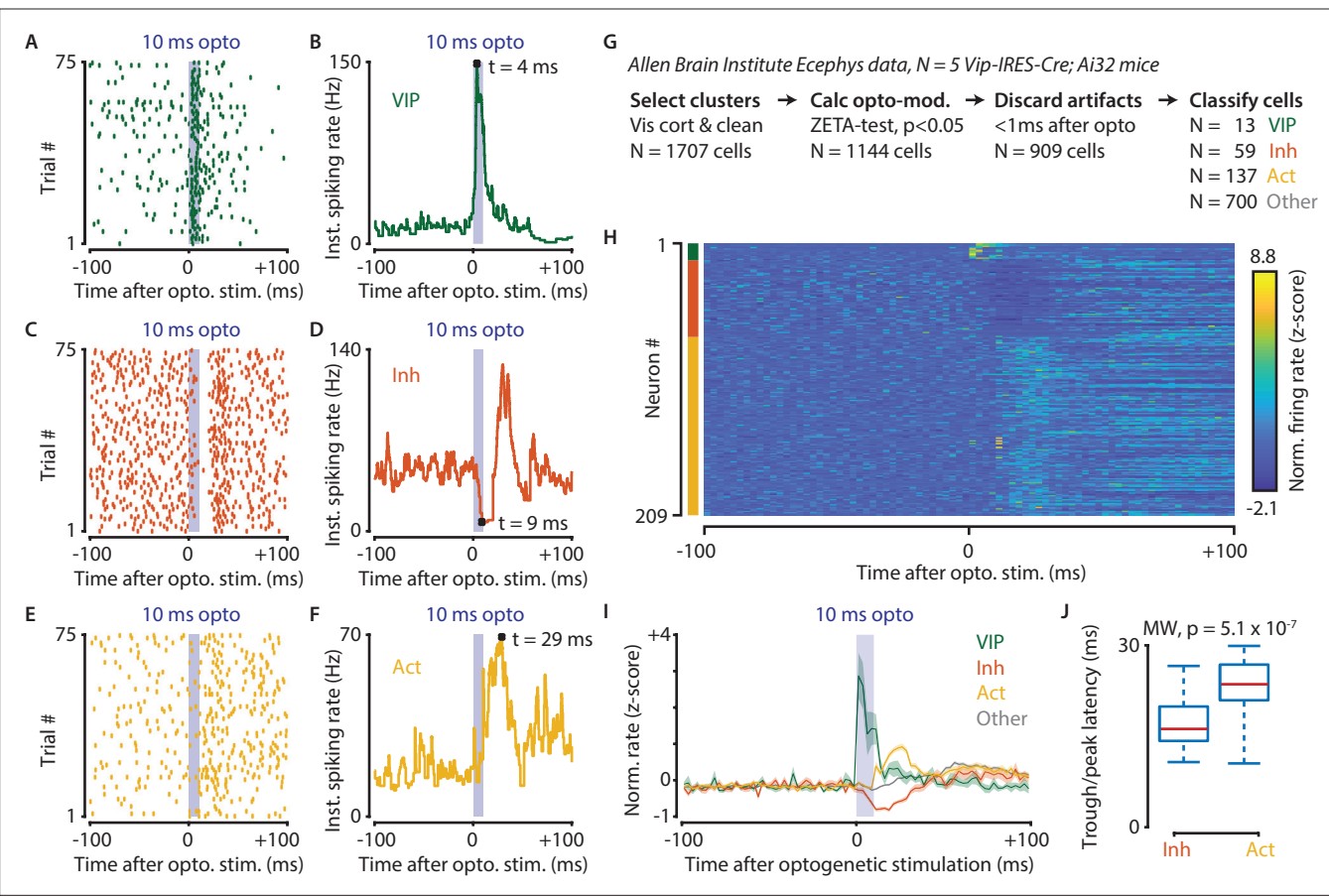

**Figure 8.** Optogenetic stimulation of VIP-expressing cells in mouse visual cortex causes short-latency inhibition and longer latency disinhibition of the local neural circuit. (**A–F**) Response of example cells classified as VIP (**A,B**), Inhibited (**C,D**) and Activated (**E,F**). (**G**) Data were recorded in visual cortex from 5 Vip-Cre mice at the Allen Brain Institute. Cells were only included if the clustering quality was sufficient (1707 cells total). A ZETA-test included cells that were modulated within (−0.5, + 0.5 s) after optogenetic stimulation (N = 1144 cells). IFR peak- and trough-latency was computed and cells were discarded if their peak was earlier than 1 ms after optogenetic stimulation onset. Remaining cells were classified as VIP (N = 13), Inhibited (N = 59), Activated (N = 137), or Other (N = 700). (**H**) Heat map showing normalized firing rate of VIP (top), Inhibited (middle), and Activated (bottom) cells. (**I**) Mean ± SEM of PSTH (2.5 ms bin size) over all VIP (green), Inhibited (orange), Activated (yellow), and Other (gray) cells. (**J**) Inhibited cells showed significantly lower mean IFR-peak latencies after optogenetic stimulation than Activated cells (Inh: 17.4 ms; Act: 23.0 ms; Mann-Whitney U-test, p = 5.1 × 10⁻⁷).

The analysis of optogenetic stimulation in visually-responsive areas is complicated by the fact that mice can see the blue light used for optogenetic stimulation. In other words, if a neuron is active after a laser pulse, it could be caused by direct stimulation, indirect circuit disinhibition, or simply be a sensory-driven response. Using the methods described in this paper to overcome these issues, we show that optogenetic stimulation of VIP-expressing cells in mouse visual cortex causes short-latency inhibition and longer latency disinhibition in separate neuronal subpopulations (*Figure 8*).

Data used for this analysis were recorded from 5 Vip-IRES-Cre; Ai32 mice at the Allen Brain Institute. Cells were only included for analysis if the clustering quality was sufficient, and they were recorded in a visual cortex area (1706 cells total in AL, AM, PM, L, V1, RL, MMP). We performed a ZETA-test to discard cells that were not modulated within the interval (–0.5, + 0.5 s) after optogenetic stimulation (N = 1144 cells remaining). Instantaneous firing rate peak- and trough-latencies were computed and cells were discarded if their peak occurred earlier than +1 ms after optogenetic stimulation onset. The remaining cells (N = 909) were classified as VIP (N = 13) if their peak latencies occurred within the 10 ms duration of the optogenetic stimulation, as Inhibited (N = 59) or Activated (N = 137) if they displayed respectively decreased or increased firing within 20 ms from the stimulus offset, or Other (N = 700). We limited our classification window to the first 30 ms after the onset of the optogenetic stimulus as it takes about 30–50 ms for retinal light responses to emerge in the visual cortex. Hence, the majority of 'Other' cells likely show sensory-driven responses.

Based on these classifications, we compared the single-cell latencies of Inhibited and Activated neurons, restricting our analysis to peaks within the 10–30 ms post-optogenetic stimulation window. The latency of Inhibited cells was significantly shorter after optogenetic stimulation than it was for Activated cells (median latency Inh = 16.5 ms, Act = 23.6, Mann-Whitney U-test, $p = 5.1 \times 10^{-7}$). These results show a VIP-mediated disinhibition mechanism operates in vivo in visual cortex, confirm slice connectivity studies (*Pfeffer et al., 2013*), and are in general agreement with results from the auditory cortex and mPFC (*Pi et al., 2013*).

## Discussion

We developed the ZETA-test, a statistical method for determining whether neuronal spiking responses are modulated by the occurrence of events, such as the onset of sensory stimulation. The ZETA-test is widely applicable: we have shown that it can accurately detect neuronal responsiveness in a wide range of settings, across various brain regions, stimuli, and recording techniques. In most cases, the ZETA-test showed markedly improved statistical sensitivity compared to established and powerful statistical techniques, such as t-tests and ANOVAs. For example, the ZETA-test detected a visual response in 42 % of the cells that were not included by a t-test and in 28 % of the cells not included by a PSTH-based ANOVA. In addition to its improved statistical performance, the ZETA-test avoids arbitrary parameter choices, as it does not require the selection of a temporal bin size. This makes the ZETA-test even easier to use than established methods, as it can be applied directly to raw spike times and stimulus onsets, and the lack of a parameter selection naturally lends itself to the bulk-analysis of large numbers of cells.

Secondly, we developed an instantaneous firing rate for analysis and visualization of neuronal firing patterns. Similar to the ZETA-test, it has two main advantages over alternative common approaches, such as PSTHs: (1) it does not require binning, and therefore removes the need to tune binning widths for individual cells or cell types and (2) its temporal resolution is only limited by the spike density, which allows more accurate determination of spike train events such as peak response latencies. When benchmarking its performance for latency-estimation, it outperformed PSTH-based approaches without the need to fine-tune bin sizes for each neuron individually (*Figure 6K*). Finally, compared to methods that rely on model fitting, such as the multiplicative inhomogeneous Markov interval (MIMI) model, the ZETA-test is considerably faster, not subject to overfitting, and does not require tuning of hyperparameters (*Figure 3—figure supplement 2*).

We investigated the performance of the ZETA-test in a variety of cases and observed that the ZETA-test sometimes performed as well as other techniques and sometimes better. So when, and why, exactly does the ZETA-test provide extra statistical power? We have shown in the methods that the ZETA-test shares mathematical properties with a Kolmogorov-Smirnov test of a neuron's spike train against permuted onset-jitter bootstraps. Various components of the ZETA-test were chosen to relax the assumptions made by other tests with the aim to gain relative invariance to the specifics of

a neuron's spike train (*Figure 3—figure supplement 3*). While the ZETA-test is slightly worse than a t-test for purely Poisson-distributed spike counts, it outperforms the t-test in every other case we tested, including those where the interspike interval distribution is highly peaked, such as in bursting cells (*Figures 1–4*). Finally, while more sophisticated model-based approaches might be able to attain better performance than the ZETA-test, this requires that their hyperparameters be individually tuned to a cell's firing statistics (e.g. MIMI-based methods; *Figure 3—figure supplement 2*). Therefore, the ZETA-test may often be the preferred choice as it required no fitting, hyperparameter tuning, and shows a statistical sensitivity superior to model-based approaches in all cases we tested.

An alternative to using the ZETA-test could be to perform a set of ANOVAs as in *Figure 5*. While this would certainly be an improvement over doing a paired t-test, multiple ANOVAs are more complex to implement and the procedure needs multiple-comparison correction (*Head et al., 2015*). More importantly, it still requires choosing an arbitrary set of bin sizes, as simply taking the optimal bin size still leads to an underperformance relative to the ZETA-test (*Figure 1G*). Moreover, the ZETA-test has the advantage of not having to choose any parameter at all. As we have shown in *Figure 8*, using the ZETA-test and IFR allowed us to significantly simplify various analysis steps when investigating the heterogeneous latency effects of optogenetic stimulation of VIP neurons.

However, like all statistical tools, the ZETA-test also has its limitations. As we designed it to be a generalist test applicable to any spike train, the ZETA-test might not perform as well as models manually fitted to describe a particular cell's response. However, we believe in many cases it will provide a superior alternative to other responsiveness tests, and response-latency determinations. The ZETA-test shows great merit especially as an unbiased parameter-free inclusion criterion, which is something that currently often varies between studies and can adversely affect the replicability of scientific results (*Mesa et al., 2021*). Another, and fairly obvious, limitation is that the ZETA-test only compares point-events (e.g. spike times) to other point-events (e.g. stimulus onsets). Consequentially, it is a powerful tool for state-of-the-art electrophysiological techniques, like Neuropixels recordings, but does not apply to calcium imaging dF/F0 traces. However, as we show in *Figure 7*, the ZETA-test performs well when applied to putative spiking events extracted from calcium imaging data. Therefore, the ZETA-test would still be useful in calcium imaging data sets where the temporal response profile of neurons to the experimental treatment is unknown. Moreover, optical recording techniques continue to improve, allowing better single-spike extractions with calcium imaging (*Packer et al., 2015*) and genetically encoded voltage indicators (*Knöpfel and Song, 2019*). As such, in all likelihood near-future neural data will remain spike-based regardless of the underlying recording technique used.

The main aim of this paper is to present the IFR and ZETA-test and show how they improve upon commonly used analysis tools, such as PSTHs, t-tests, Markov-model-based approaches, and ANOVAs. However, we also show some results that by themselves are scientifically noteworthy. For example, our results indicate that almost all V1 neurons ( > 93%) are responsive to drifting gratings, even if the spatial frequency (SF) and temporal frequency (TF) parameters are not optimal to drive these cells (*Figure 3*). This calls into question the idea that V1 cells are generally narrowly tuned and can only be driven by specific stimulus feature parameters (*Xing et al., 2004*). Our analysis indicates that neuronal responsiveness rarely drops to 0 for a combination of stimulus features, as otherwise we would not have found such a large number of responsive V1 cells using a feature-sparse stimulus (24 directions, 1 SF: 0.05cpd, 1 TF: 2 Hz).

This suggests that the response of V1 neurons to features such as SF and TF may drop off rather slowly with distance to their preferred stimulus properties. Conceptually, this would mean that the V1 neural code may in fact be more 'dense' than proposed by some (*Ohiorhenuan et al., 2010*; *Olshausen and Field, 1997*; *Vinje and Gallant, 2000*). That said, our results show that only a relatively small group of neurons is strongly driven by any one stimulus: sparse coding may therefore still operate in V1, but it acts on top of a dense, but weak code.

Two other noteworthy findings are that VIP cell activation drives disinhibition in visual cortex (*Figure 8*) and that the strength of visuomotor signals and modulation by spatial location are negatively correlated (*Figure 7*). This suggests that bottom-up and top-down processing in visual cortex may be mediated by distinct neuronal subnetworks. While this functional segregation has previously been shown based on a macroanatomical (e.g. laminar) analysis (*Kok et al., 2016*; *Markov et al., 2014*; *Poort et al., 2012*; *Self et al., 2019*), our results suggest that this functional segregation also holds at a microanatomical level for neurons located within a single recording plane (*Figure 7*).

In conclusion, the IFR and ZETA-test are simpler, more statistically powerful, and less error-prone tools than bin-based PSTHs, t-tests, Markov-model based approaches, and ANOVAs widely used in neuroscience today. Statistically underpowered studies are still common in neuroscience, which makes the development of statistically sensitive tools especially important (*Button et al., 2013*). Moreover, when power analyses are used to determine the necessary sample size, then increased statistical sensitivity, such as with the ZETA-test, can reduce the number of required experimental animals. To facilitate the adoption of the ZETA-test by the neuroscientific community, we provide easy-to-use and well-documented open-source implementations online in MATLAB (https://github.com/JorritMontijn/ZETA, *Jorrit, 2021b*) and Python (https://github.com/JorritMontijn/zetapy).

## Materials and methods
### ZETA

Well documented and easy-to-use Matlab and python code performing the procedures described in the following paragraphs can be found here: https://github.com/JorritMontijn/ZETA and https://github.com/JorritMontijn/zetapy.

We developed a timescale-free, binning-less statistical test for determining whether a neuron shows a time-locked modulation of spiking activity. It is derived from a metric that represents the reliability, as number of standard deviations away from chance, that the temporal density of spikes is non-random across trial repetitions. This metric, we call ZETA ($\zeta$), can be computed on a vector of $i = [1 \ldots N]$ spike times $\boldsymbol{x}$, and a vector of $k = [1 \ldots q]$ event times (e.g. stimulus onsets), $\boldsymbol{w}$, using the following steps.

First, we make a vector $\boldsymbol{v}$ of the spike times in $\boldsymbol{x}$ relative to the most recent stimulus onset, as when making a raster plot of spike times:

$$v_i = x_i - w_k \tag{1}$$

where

$$w_k < x_i \leq w_{k+1} \tag{2}$$

Next, we remove all spike times that are larger than a cut-off value $\tau$, for example the trial duration, and add two artificial spikes at t = 0 and t=$\tau$ to ensure coverage of the full epoch. We sort the $n$ spike times in $\boldsymbol{v}$ such that $v_i < v_{i+1}$, and calculate the fractional position $g_i$, ranging from $1/n$ to 1, of each spike time in $\boldsymbol{v}$:

$$g_i = i/n \tag{3}$$

Therefore, another interpretation is that $\boldsymbol{g}$ represents a neuron's cumulative density function sampled at the spike times in $\boldsymbol{v}$. In order to quantify whether this distribution is different from our null hypothesis – that is that the neuron's firing rate is not modulated with respect to the stimulus onset – we compare this vector to a linear baseline density vector $\boldsymbol{b}$. If a neuron's spiking rate is constant, the cumulative density function is linear over time, and therefore the expected fractional position of spike $i$ at time $v_i$ converges to the spike time divided by the trial duration $\tau$ as the number of events $q$ increases:

$$\lim_{q \to \infty} b_i = v_i/\tau \tag{4}$$

The difference $\delta_i$ between $g_i$ and $b_i$ therefore gives a neuron's deviation from a temporally non-modulated spiking rate at time point $v_i$:

$$\delta_i = g_i - b_i \tag{5}$$

As we show in the Materials and methods section 'A proof of time-invariance', using $\delta_i$ to compute ZETA would make it dependent on the choice of onset times. Therefore, we create $\boldsymbol{d}$, a time-invariant mean-normalized version of $\delta$:

$$d_i = \delta_i - \bar{\delta} \tag{6}$$

where

$$\bar{\delta} = \frac{1}{n} \sum_{i=1}^{n} \delta_i \tag{7}$$

We then define the Zenith of Event-based Time-locked Anomalies (ZETA, or $\zeta_r$) as the most extreme value, that is the maximum of the absolute values:

$$\zeta_r \equiv \max\left(\left|\boldsymbol{d}\right|\right) \tag{8}$$

## Null hypothesis for ZETA

Having calculated ZETA from the temporal deviation vector $\boldsymbol{d}$, we wish to quantify its statistical significance. First, we scale it such that its value is interpretable as a z-score. We therefore construct a null hypothesis distribution by repeating the above procedure $P$ times with jittered event-times $\boldsymbol{w'}$, where we move each event time by a random sample drawn from the interval [$-\tau$, $\tau$]. This way, we calculate the chance of observing randomly high values in $\boldsymbol{d}$ without having to make assumptions about the underlying distribution of $\boldsymbol{d}$. However, a naive approach would lead to difficulties here, as jittering $\boldsymbol{w}$ also changes the corresponding values of $\boldsymbol{v}$; and any jittered vector $\boldsymbol{d'}$ we obtain would be sampled at different times than the original vector $\boldsymbol{d}$. Therefore, we instead linearly interpolate the values of jittered fractional position vector $\boldsymbol{g'}$ at the original spike times of $\boldsymbol{v}$. First, we construct a vector $\boldsymbol{f}$ of fractional spiking positions analogously to $\boldsymbol{g}$, but based on jittered event-times $\boldsymbol{w'}$:

$$f_i = i/n' \tag{9}$$

Note that we cannot simply take $\boldsymbol{g}$, as the total number of spikes $n'$ in this jittered version is likely different from the original number of spikes $n$, because we only consider the spike times in the interval [0, $\tau$] after the (jittered) event times. Next, we interpolate the values of $\boldsymbol{f}$ at sample times $\boldsymbol{v'}$ to the original sample times $\boldsymbol{v}$:

$$g'_i = (1-w)f_{j-1} + wf_j \tag{10}$$

where

$$w = \frac{v'_i - v_j}{v_k - v_j} \tag{11}$$

with

$$v_{j-1} \leq v'_i \leq v_j \tag{12}$$

We repeat this process $P$ times; where for each jitter iteration $j$, we calculate $\boldsymbol{\delta'}(j)$:

$$\boldsymbol{\delta'}(j) = \boldsymbol{g'}(j) - \boldsymbol{b} \tag{13}$$

Note that $\boldsymbol{b}$ is invariant with respect to the jitter iteration, as it is simply the $n$-element linear vector from $1/n$ to $n$. As before, we mean-normalize $\boldsymbol{\delta'}(j)$ to obtain a temporal deviation vector $\boldsymbol{d'}(j)$.

$$\boldsymbol{d'}(j) = \boldsymbol{\delta'}(j) - \bar{\delta'}(j) \tag{14}$$

Now we can define a null-hypothesis ZETA sample $j$ as:

$$\zeta'(j) \equiv \max\left(\left|\boldsymbol{d}'(j)\right|\right)$$

(15)

## Statistical significance of ZETA

Having constructed a way to generate samples from a null-hypothesis distribution, we are left with the task of using it in calculating the statistical significance of ZETA. If we had infinite samples, we could directly calculate the percentile of the empirical $\zeta_r$ from the null-distribution. However, as this is computationally intractable, we will approximate the true distribution from a finite number of null-hypothesis samples. From extreme value theory we know that the distribution of maximum values is known as a Gumbel distribution (**Gumbel, 1941**). Its cumulative density is given by:

$$F\left(x; m, \beta\right) = e^{-e^{-(x-m)/\beta}}$$

(16)

Here, $x$ is the sample maximum (i.e., $\zeta_r$), $m$ is the mode, and $\beta$ is the scale parameter. Therefore, we need to find $m$ and $\beta$, which can be derived from the estimated sample mean and variance over jittered ZETAs of $\zeta'$. The mean $x$ and variance $v$ are given by **Gumbel, 1954**:

$$\bar{x} = m + \beta\gamma$$

(17)

$$v = \frac{\pi^2\beta^2}{6}$$

(18)

Here, $\gamma$ is the Euler–Mascheroni constant ($\gamma \approx 0.577$), $m$ is the mode, and $\beta$ is the scale parameter. Using $v = \mathrm{Var}\left(\zeta'\right)$, and **Equation 18**, we can write the scale parameter $\beta$ as:

$$\beta = \frac{\sqrt{6 \cdot Var(\zeta')}}{\pi}$$

(19)

Then using **Equation 17** and $\bar{x} = \bar{\zeta}'$, the mode can be computed from $\beta$ and the mean:

$$m = \bar{\zeta}' - \beta\gamma$$

(20)

Now we can define the p-value by reading out the cumulative Gumbel distribution at $\zeta_r$:

$$p = 1 - F\left(\zeta_r; m, \beta\right)$$

(21)

Finally, we can use $p$ with the standard normal's quantile function $\Phi^{-1}$ to obtain a corrected ZETA $\zeta$ that is interpretable as a z-score:

$$\zeta = \Phi^{-1}\left(1 - \frac{p}{2}\right)$$

(22)

Note that when we refer to ZETA or $\zeta$ in the rest of the manuscript, we mean the corrected version and its p-value as defined above.

## Computing an optimal bin size

For the analyses where we used an optimal binning width to compare the performance of the ZETA-test and a bin-wise ANOVA, we computed the optimal bin width using the procedure described by **Shimazaki and Shinomoto, 2007**. Their method describes a loss function that can be computed for a given bin-width. To find the optimal bin width, we used a simple iterative 10-point grid search until a local minimum was found. The code used for finding the optimal bin size is available online at https://github.com/JorritMontijn/GeneralAnalysis, (copy archived at swh:1:rev:7f866e0c875af17e9d76fdfbd-8cec3d41145c031, **Jorrit, 2021a**) in the function *opthist.m*.

### The multiplicative inhomogeneous Markov Interval (MIMI) Model

A classic model for neuronal firing rates are inhomogeneous Poisson processes, where a time-dependent function *f(t)* can be used to describe how the mean firing rate $\lambda$ of a cell varies with time *t* after some experimental intervention, such as the onset of a visual stimulus (**Kass et al., 2014**):

$$\lambda\left(t\right) = f\left(t\right) \tag{23}$$

Spike-times $\boldsymbol{v}$ can then be generated by sampling from an exponential distribution with an inter-spike interval equal to $1/\lambda$:

$$v_{i+1} = v_i + \mathrm{Exp}\left(\tfrac{1}{\lambda}\right) \tag{24}$$

While this framework is attractive in its simplicity, it cannot capture several important properties of spiking dynamics, such as refractory periods or burst firing. Even when one is only interested in the question whether a particular neuron responds to a visual stimulus, bursting cells might produce apparent bumps in a peri-stimulus time histogram (PSTH). This could lead to the possibly erroneous conclusion that a cell is stimulus-modulated, simply because their spiking patterns are non-Poisson by nature. This problem is remedied by a class of models that combine a Poisson process with a renewal process, which describes the likelihood of a spike conditioned on the time since the last spike. These processes are called multiplicative inhomogeneous Markov interval (MIMI) processes (***Kass and Ventura, 2001***):

$$\lambda\left(t, t - s_*\left(t\right)\right) = \lambda_1\left(t\right) \cdot \lambda_2\left(t - s_*\left(t\right)\right) \tag{25}$$

Here, $s_*(t)$ is the time of the last spike, the $\lambda_1$ term refers to the inhomogeneous Poisson process described above, while the latter $\lambda_2$ term captures the inter-spike-interval dependent spiking probability. While this model can be extended to include interaction terms, n-back spike dependencies, and bias constants per trial, prior work has shown these additions do not appreciably improve the fitting quality (***Kass and Ventura, 2001***). As an additional baseline model, we therefore also compared the performance of the ZETA-test to that of a method based on the MIMI-model (***Equation 25***).

## MIMI-model fit evaluation as a statistical test for responsiveness

To use the MIMI-model framework as a statistical test, we binned spikes in 1 ms bins. We used cubic splines with 16 B-form coefficients spread uniformly over the trial's 1.5 s duration for the inhomogeneous Poisson component, and 16 B-form coefficients spread uniformly over a time horizon of 500 ms for the renewal-process component (***Kass et al., 2014***). We then simultaneously fitted these 32 coefficients to produce the closest match to the neuron's spike train by running a least-squares curve fitting algorithm. Specifically, we minimized the error between the stimulus-locked trace reconstituted from the B-form splines and the real average spiking rate per bin of the PSTH. The fitting procedure used 1 ms binning, but the resulting model can be resolved at theoretically infinitesimal time steps. We ran a couple of fits with different numbers of coefficients ranging from 8 + 8–32 + 32, but this did not strongly impact either the fitting quality or computation time (***Figure 3—figure supplement 2***).

Theoretically, we could now follow the same procedure as with the ZETA-test by generating a null-hypothesis distribution from multiple iterations of jittered onset times. However, the MIMI fitting procedure's computational time cost meant this was not a realistic solution: even a single MIMI-model fit took 557 times as long to run as the ZETA-test, which already included 100 random jitter iterations (***Figure 3—figure supplement 2***, panel H). This means that using 100 random jitters for the MIMI-model test would take 55,700 times as long as the ZETA-test. We therefore took an alternative approach by calculating pairwise d' values (i.e. the distance in standard deviations between the two bins) for all bin pairs, and transformed the highest d' into a p-value. This is obviously a suboptimal approach, as this will lead to a high number of false positives. However, our goal here is not to develop a full-fledged MIMI-based test, but rather to verify what the statistical sensitivity of such an approach could be. It is possible to investigate the statistical sensitivity as the low p-value bias exists for both real data and shuffle controls, so instead of comparing the inclusion level at an alpha of 0.05, we performed an ROC analysis over all cells in V1.

The ROC analysis is insensitive to the absolute level of significance values, but instead provides insight in the discriminability of real inclusions from false positives (***Figure 3—figure supplement 2***, panel G). To keep the computational time tractable, we subsampled the data to include only V1 cells, and only their response to the first 480 drifting grating trials. We compared the ZETA-test, t-test and MIMI-model method as described above. The ZETA-test gave an area under the curve (AUC) of 0.996, the t-test 0.900, and the MIMI-method 0.749. However, we noticed that the MIMI-method

appeared to fail mostly for cells with low firing rates, so we also added a hypothetical curve where we only included neurons with > 1000 spikes during the 480-trial long epoch (MIMI-1k). This significantly boosted the MIMI-method's discriminability to an AUC similar to the t-test's at 0.898. Perhaps choosing a different number of coefficients would improve the MIMI test, but the issue remains that this test does not work without manual tuning. These results suggest it will require significant work to develop a MIMI-based test that can compete with a t-test, and that even if we were successful, it might not exceed the ZETA-test's performance. Moreover, as the MIMI-model requires parameters to be iteratively fitted to experimental data, it is multiple orders of magnitude slower than the ZETA-test and t-test. To conclude, full MIMI-model based methods do not seem to be suited for unsupervised, large-scale use as neuronal responsiveness tests.

## Decomposition of ZETA

While using full multiplicative inhomogeneous Markov interval models of the form of *Equation 25* is not a viable option when creating a responsiveness test, we used it as a starting point to further explore which properties of the ZETA allow it to function so well. As already noted above, the main problem for developing a robust responsiveness test is finding a suitable null-hypothesis distribution to test against. As a first naïve baseline, we built a simple test that checks whether a cell's firing rate, binned with width $\tau$ and averaged across trials, differs from a homogeneous Poisson process with rate $\lambda$. Under this null-hypothesis, the number of spikes per bin $X$ is therefore distributed as:

$$X \sim \text{Pois}\left(\lambda\tau\right) \tag{26}$$

The null-hypothesis random variable $H_0$ for rates averaged over T trials follows:

$$H_0 = 1/T \sum_{i=1}^{T} X_i \tag{27}$$

Whether a neuron's observed number of spikes per bin, averaged over trials, differs from this null distribution can then be tested using a standard Kolmogorov-Smirnov test.

We benchmarked this approach with V1 cells, using 1 ms bins, and performed an ROC analysis of real inclusions versus false positives using two different ways to construct neuronal responses that are unmodulated by the visual stimulation: (1) we shuffled the inter-spike intervals to construct a new set of spike times (*Figure 3—figure supplement 3A*) and (2) we jittered the stimulus onset times by ±6 s (*Figure 3—figure supplement 3B*). We found that the Poisson-based test performed well if the H0 spiking statistics matched the shuffle-control (Poiss-KS, AUC = 0.976), but failed due to high numbers of false positives when we instead jittered the stimulus onset times (Poiss-KS, AUC = 0.611). By construction, the fluctuations in the rate are not linked to the stimulus onset times anymore, but the distribution of the spike times over the trial period are not consistent with a Poisson neuron.

This may seem a trivial result, as the null-hypothesis distribution in this latter case does not match the null (Poisson) distribution used by the statistical test. This is a critical issue, however: in the case of real experimental data sets, there is no known ground truth, so a robust statistical test for neuronal responsiveness must be able to handle a wide variety of intrinsic spiking behaviors. Clearly, the Poisson test fails this requirement, as it is only able to distinguish i.i.d. Poisson-distributed spiking from anything that is not exactly i.i.d. Poisson-distributed spiking. For comparison, both the ZETA-test and t-test show robust behavior that is insensitive to the specifics of the shuffle-control we use. The ZETA-test gives an AUC of 0.983 using ISI-shuffles and 0.984 using onset jittering; and the mean-rate t-test gives AUCs of 0.899 and 0.902 respectively. In the following section, we investigate which properties of the ZETA-test allow it to perform so much better than the Poisson test.

The first aspect we investigated is the assumption of the homogeneous Poisson-distributed spiking when a neuron is unmodulated by visual stimulation. We know that neurons can be intrinsically bursting and have refractory periods, so even the most purely sensory-driven cell in V1 is likely to not fire i.i.d. Poisson when no visual stimulus is present. A more versatile and possibly more accurate H0 might be a renewal process. We therefore constructed an inter-spike-interval (ISI)-based test, where we first calculated a neuron's inter-spike intervals $d\textbf{\textit{t}}$ from its spike time vector $\textbf{\textit{t}}$:

$$dt_i = t_i - t_{i-1} \tag{28}$$

We then randomly permuted the ISIs, creating a shuffled ISI vector $d\boldsymbol{t}^s$ and using it to construct a null-hypothesis vector of spike times $\boldsymbol{t}^0$:

$$t_i^0 = t_{i-1}^0 + dt_i^s \tag{29}$$

In effect, this null-hypothesis vector is a different random sample of the same renewal process that would generate the real neuron's spiking times, under the simplifying assumption that the spike times are generated by a renewal process. We constructed mean firing rates $\boldsymbol{x}^0$ by binning $\boldsymbol{t}^0$ using 1 ms bins. We repeated this 100 times, and used a two-sample K-S test between the real binned spike count vector $\boldsymbol{x}$ and shuffle-control spike count matrix $\boldsymbol{X}^0$, as we also did for the Poisson test described above. Unfortunately, this SISI-KS (Shuffled Inter-Spike-Interval Kolmogorov-Smirnov) test performed very similar to the Poisson test, resulting in an AUC of 0.985 for the ISI-shuffle control, and an AUC of 0.610 for the jitter-control.

This similarity might be explained by the ability of the KS test to distinguish with high sensitivity between the spiking distributions obtained from shuffling ISIs and jittering stimulus onsets, as these are not identical. If we wish to construct a more robust test, we must therefore use a procedure that is less sensitive to the full shape of the H0 distribution, and only takes into account the likelihood of observing extreme deviations from the average firing rate. This will make the test less sensitive to real stimulus-induced activity, but also less sensitive to errors in the specific shape of the null hypothesis distribution we use to estimate the neuron's natural variability. We achieved this by using the Gumbel-distribution of maximum absolute deviations in our random ISI-shuffle samples to calculate a p-value of the maximum absolute deviation in the real, unshuffled, PSTH. The procedure works as intended: this SISI-G test gives an AUC of 0.844 for the ISI-shuffle controls, and an AUC of 0.821 for the jitter controls.

We noticed that the SISI-G test suffered from high variance in the PSTH when using 1 ms bins, especially for cells with few spikes. One option would therefore be to increase the bin width, but this would come at the expense of temporal resolution and ability to detect short peaks of activity. We therefore opted instead to calculate the maximum absolute deviation of the cumulative sum of spikes counts, similar to the ZETA-test's, but in this case over the discrete 1 ms spike count vector $\boldsymbol{x}^0$. For the SISI-∫G test, we defined the normalized cumulative spike count vector T0 (similar to *Equation 29*) as:

$$T_j^0 = \sum_{i=1}^{j} x_i^0 - \bar{x}^0 \tag{30}$$

Moreover, as this would create fixed points with 0 variance at j = 1 and j = n, we also mean-subtracted the T0 vector itself:

$$\boldsymbol{T}^{0,c} = \boldsymbol{T}^0 - \bar{T}^0 \tag{31}$$

In essence, this test is a 1 ms binned and ISI-shuffle based version of the binless onset-jittering ZETA-test. Benchmarking this test, we found that it performed close to, but slightly less well than, the ZETA-test. This SISI-∫G test gave an AUC of 0.968 for the ISI-shuffle controls and an AUC of 0.974 for the jitter controls.

Finally, we created an alternative, also binless, version of the ZETA-test where we created the null distributions by shuffling the inter-spike intervals rather than jittering the stimulus onsets. The alternative ZETA-ISI test performed at a level indistinguishable from ZETA; the ZETA-ISI gave an AUC of 0.986 for the ISI-shuffle controls and an AUC of 0.982 for the jitter controls. To conclude; the (alternative) ZETA-test strongly outperforms other tests, mainly for two reasons: (1) using the Gumbel distribution to calculate a cell's significance based on the most extreme stimulus-locked spiking deviation rather than a KS test allows the ZETA-test to be relatively invariant to the full, and a priori unknown, spike time distribution of a neuron and (2) using an integral-based approach has a timescale-free smoothing effect that reduces spurious peaks that can occur in the firing rate domain. Finally, the binless ZETA-test shows a small, but significant, improvement over its 1 ms binned cousin in terms of both statistical power and computational efficiency (*Figure 3—figure supplement 3C*). While the random null distribution (ISI shuffling or onset jittering) did not seem to have a large impact on the ZETA-test's performance for this data set of predominantly regular-spiking V1 neurons, the following section shows that this distinction becomes more important when one tests the responsiveness of bursting cells.

## Simulated bursting cells: onset jittering versus inter-spike interval shuffling

Stimulus onset jittering and inter-spike interval shuffling produce different distributions, unless a neuron's probability of spiking only depends on the time since the last spike. The previous section showed that, when using our V1 data set, performance has already saturated too much to show a difference between the ISI-shuffle ZETA and onset-jitter ZETA tests. To better differentiate their performance, we therefore generated a population of simulated bursting cells. While bursting cells are rare in visual cortex, they are abundant in many brain regions, such as the subiculum and others (*Cooper, 2002*; *Mattia et al., 1993*). To test the performance on bursting cells, we generated artificial spike trains, using the parameters for burst properties from *Chen et al., 2009* as listed in *Table 1*.

All cells were assigned a background single-spike firing rate that was on average 1 Hz; on top of these single spikes we generated bursts with varying length and inter-spike intervals. The inter-burst intervals were 16 s on average during inter-trial intervals and non-preferred stimuli; and 0.8 s during the neuron's preferred stimulus. Each neuron had a randomly assigned preferred orientation and tuning width following a von Mises distribution. We presented 20 repetitions of 24 orientations (15 degree steps) for a total of 480 trials. We generated responses for 1,000 independently parameterized neurons (see *Figure 3—figure supplement 3D-E*). To create stimulus non-responsive neurons, we set the inter-burst intervals to be approximately 6.8 s both within and outside stimulus presentation; we chose this value to ensure the spiking rate distributions for stimulus-responsive and unresponsive neurons were approximately equal (25th-75th population percentiles of firing rates; stimulus-responsive: 6.7–19.7 Hz; unresponsive: 6.9–20.7 Hz).

Running the same benchmark as before on this artificial data set, we found that the onset-jitter ZETA-test indeed outperformed both the ISI-shuffle ZETA-test and mean-rate t-test (*Figure 3—figure supplement 3F*). The ZETA-test gave an AUC of 0.941, the ISI-shuffle ZETA-test an AUC of 0.845, and the mean-rate t-test an AUC of 0.902. This difference in performance can be attributed to the fact that jittering stimulus onsets keeps the properties of individual bursts intact, while ISI-shuffling changes these properties, leading to more variable burst spike trains. While shuffling of ISIs or stimulus onsets both produce spike trains that are unmodulated by stimulus presence, we have shown that they are not equivalent.

## Multi-scale derivatives of ZETA for latency detection

The ZETA-test indicates whether a neuron shows reliable deviations in spiking rate with respect to a particular series of events. The time of this maximum deviation, however, is not necessarily when the neuron shows its strongest firing rate modulation, but rather when the cumulative distribution reaches peak statistical significance. Therefore, in order to use the ZETA procedure to calculate the time of peaks in modulations (e.g. onset latencies), we should take the derivative of the temporal deviation vector $d$ underlying ZETA. A naïve approach with a simple spike-to-spike derivative unfortunately yields a curve with many spurious peaks. One solution would be to calculate the derivative over a larger time interval, but this comes at the expense of temporal resolution. Moreover, many different cell types exist with different dominant time constants. To balance temporal resolution and robustness, we therefore developed a multi-scale derivative procedure. First, we define a vector $t$ of $S$ timescales at which to compute derivatives. By default, we define the timescales to lie on a logarithmic scale with base 1.5, as this gave a reasonable trade-off between computational speed and accuracy. Base values closer to one will give more accurate results at the cost of computational speed. For base $b$ and a trial duration of $\tau$:

$$t = b^p \tag{32}$$

where

$$p = \left\{ x \mid x \in \mathbb{Z}, log_b 10^{-3} < x < log_b \tfrac{\tau}{10} \right\} \tag{33}$$

The derivative $\dot{d}$ at spike $i$ can then be defined for $d_i$ and timescale $t_k$ as:

$$\dot{d}_{i,k} = \frac{d_b - d_a}{v_b - v_a} \tag{34}$$

where

$$a = \arg \max v_a \left\{ v_a \mid v_a \in \boldsymbol{v}, v_a < v_i - \tfrac{t_k}{2} \right\}$$
$$b = \arg \min v_b \left\{ v_b \mid v_b \in \boldsymbol{v}, v_b > v_i + \tfrac{t_k}{2} \right\} \tag{35}$$

Here, $\boldsymbol{v}$ are spike times, following the definition above. To avoid undefined edges, we set $a$ and $b$ to one and $n$ respectively, iff $v_i \pm t_k/2$ falls outside the interval $[0, \tau]$. Taking the mean over all $S$ timescales, we obtain an average of multi-scale derivatives $\boldsymbol{m}$,

$$m_i = \tfrac{1}{S} \sum_{k=1}^{S} \dot{d}_{i,k} \tag{36}$$

which has two important properties. First, long-timescale derivatives tend to 0, so there is a bias of $\boldsymbol{m}$ to more strongly follow shorter timescales. Secondly, random noise at the shortest timescales averages out over multiple short-timescale derivatives. Therefore, these two properties combined lead $\boldsymbol{m}$ to reflect the shortest timescales at which a real signal starts to emerge from random noise.

## Calculation of high-resolution instantaneous firing rates

Another interesting property of the mean multi-scale derivative $\boldsymbol{m}$ is that it scales with the actual firing rate. In other words, it produces a time-locked neural activation curve, similar to a peri-stimulus time histogram (PSTH). If we properly rescale $\boldsymbol{m}$, we can therefore create an instantaneous firing rate metric with a temporal resolution that is only limited by the spike density.

Remember that the temporal deviation vector $\boldsymbol{\delta}$ itself is scaled to lie between −1 and +1, as its value depends on the difference between the fractional position of a spike (from 0 to 1) and the linear interval from $x,y=[0,0]$ to $[\tau,1]$. The theoretical lower limit of the multi-scale derivative is therefore $-1/\tau$. This can be illustrated as follows. Imagine a hypothetical neuron where all spikes are fired in an arbitrarily short interval close the start of each trial. This means that $\boldsymbol{\delta}$ rises from 0 to 1 in a short interval and from then decays linearly from 1 to 0 over an interval of $\tau$. As $\dot{d}$ is defined between two spikes (including window edges 0 and $\tau$), this means that the lowest possible value in this extreme case is the point between the last spike $n$ and $\tau$, i.e.:

$$\dot{d}_{\min} = \frac{d_\tau - d_n}{v_\tau - v_n} = \frac{0-1}{\tau-0} = -1/\tau \tag{37}$$

The lowest possible firing rate is 0 Hz, which therefore corresponds to $-1/\tau$. An upper bound for $\dot{d}$ does not exist, as an arbitrarily short interval with a finite number of $n$ spikes would lead to arbitrarily high $\dot{d}$ :

$$\dot{d}_{\max} = \frac{d_j - d_i}{v_i - v_j} = \frac{1/n}{t - (t+1/\infty)} = \infty \tag{38}$$

This is a desirable property, as the maximum instantaneous firing rate of a neuron is not theoretically constrained. Finally, the average firing rate of our metric should correspond to the real average firing rate in Hz ($n/\tau q$), where $q$ is the number of events as defined above. We therefore define our instantaneous firing rate metric $r$ as:

$$r = \frac{n}{\tau \cdot q} \cdot \left( \frac{\boldsymbol{m} + 1/\tau}{\bar{m} + 1/\tau} \right) \tag{39}$$

Here, $\bar{m}$ is the weighted average of $\boldsymbol{m}$ by the inter-spike interval, such that the averaging occurs in the time domain and not the spike-number domain:

$$\bar{m} = \tfrac{1}{\tau} \sum_{i=2}^{n} \frac{m_{i-1} + m_i}{2} \left( v_{i-1} - v_i \right) \tag{40}$$

Considering the definitions above, we can therefore state that the maximum firing rate in $\boldsymbol{r}$ occurs where

$$r_i = \tfrac{1}{S} \sum_{k=1}^{S} \frac{b_k - a_k}{v_{b_k} - v_{a_k}} \tag{41}$$

is at its maximum, with $a_k$ the index of the largest spike time smaller than $v_i - \frac{t_k}{2}$ and $b_k$ the index of the smallest spike time larger than $v_i + \frac{t_k}{2}$; with $t_k$ being the logarithmically distributed time ranges. In the limit of a high number of spikes, we find that $v_{a_k} \approx v_i - \frac{t_k}{2}$ and $v_{b_k} \approx v_i + \frac{t_k}{2}$, and therefore,

$$r_i = \frac{1}{S} \sum_{k=1}^{S} \frac{\#\text{spikes in}\left(v_i - \frac{t_k}{2}, v_i + \frac{t_k}{2}\right)}{t_k} \tag{42}$$

which is an average of the instantaneous spike rates at $v_i$ at timescales $t_k$.

## Mean-rate artificial Poisson neurons

We tested whether ZETA required short bursts of activity to work by generating artificial spike trains that only varied in mean-rate between a 1 s stimulus presentation and a 1 s inter-stimulus interval, and did not show onset peak responses. We created spike trains for 100 neurons with an orientation preference $\theta$ randomly sampled from a uniform distribution on the interval $(0,\pi)$. The shape of the tuning curve was defined as the sum of two von Mises distributions centered at preferred orientation $\theta$ and $\theta + \pi$ with a concentration parameter of $\kappa = 5 + \varepsilon$, where $\varepsilon$ was randomly sampled from a uniform distribution on the interval $(0,5)$. The von Mises probability density function with mean $\theta$ and concentration parameter $\kappa$ is given by:

$$f\left(x \mid \theta, \kappa\right) = \frac{e^{\kappa \cos(x-\theta)}}{2\pi I_0(\kappa)} \tag{43}$$

Here, where $I_0(\kappa)$ is the modified Bessel function of order 0. The baseline mean spiking rate $\mu_{base}$ was defined by randomly sampling from an exponential distribution with a mean of $\lambda_{base} = 5$ Hz. $\mu_{base}$ defined the trough of the neuron's tuning curve (i.e. the activity at $\theta + \pi/2$ and $\theta - \pi/2$) as well as the activity of the neuron when no stimulus was present. The firing rate for the preferred stimulus $\mu_{stim}$ was determined by similarly sampling from an exponential distribution with a mean of $\lambda_{stim} = \mu_{base} + 20$ Hz. We generated spiking activity for 160 trials (20 repetitions of 8 stimulus orientations: $\theta_{stim} = [0, 45, \ldots, 315]$). The average baseline firing across all n = 10,000 artificial neurons was therefore 5 Hz and the average firing rate during the preferred stimulus was 25 Hz. Spike times were generated for each trial-epoch (stimulus/baseline) independently by consecutively drawing inter-spike intervals from a Poisson distribution with $\lambda = 1/\mu$.

## Artificial Poisson neurons for peak-latency benchmarking

We also addressed the question whether our instantaneous firing rate metric was sufficiently robust to allow accurate peak-time detection over a range of background firing rates and a range of peak widths. The procedure here was similar as above, with the exception that the firing rate during baseline and stimulus periods was identical: $\mu_{base} = \mu_{stim}$. We tested 13 base rates (0.5–32 Hz) and 19 jitter widths (1–10 ms in steps of 0.5). For each combination of base rate and jitter width, we generated 100 neurons and 100 trials per neuron. Peaks were added to the background activity by adding a single spike to half of all trials. Spike times were chosen by random sampling from a normal distribution with the standard deviation equal to the above jitter width and centered at 100 ± 10 ms after stimulus onset.

Figure panels 6 J,K used slightly different parameters. Instead, we used 10 jitter widths (1–10 ms in steps of 1), simulated only a base rate of 32 Hz, used 160 trials for 1000 neurons, and compared the peak-latency detection using the ZETA-IFR with binning windows ranging from 1.00 ms to 57.67 ms; a logarithmic scale of base 1.5 with the exponent ranging from 0 to 10 in steps of 1.

## Acquisition and preprocessing of laminar probe data (neuronexus and neuropixels)

We performed silicon probe recordings in six C57BL/6 mice, 2–7 months of age. Mice were housed in a 12 hr/12 hr dark/light cycle with ad libitum access to food and water. All experiments were approved by the animal ethics committee of the Royal Netherlands Academy of Arts and Sciences, in compliance with all relevant ethical regulations.

Mice were habituated for 1–4 weeks before being implanted with a cranial bar used for head-fixation. Mice were anesthetized with isoflurane (3 % induction, 1–1.5% maintenance in 50 % O2) and injected subcutaneously with an analgesic and anti-inflammatory compound (Metacam, 2 mg/kg).

The eyes were protected from drying by Cavasan eye ointment. They were moved to a stereotact with a thermal mat to keep their core temperature at 37 °C, and the fur on their heads was removed. Once anesthesia was sufficiently deep, as indicated by the absence of a toe-pinch reflex, we applied lidocaine locally on the skin of the head and sterilized it with 70%-ethanol or betadine. The skin was removed, the skull cleaned, and a small metal rod was fixed to the skull anterior of bregma with the use of blue-light curing dental cement. If necessary, we sutured the skin, and let the mice recover for 2–7 days.

After the mice recovered as indicated by a return to their pre-operative weight, mice were habituated to sitting head-fixed in the electrophysiology rig for 3–10 days. Once habituated, they underwent a craniotomy surgery, following the same preparatory steps as described above. Before performing the craniotomy (1.5–3 mm in diameter), we first constructed a small ring of dental cement so the brain could be bathed in saline during recordings to avoid tissue desiccation. Once the craniotomy was complete, this ring was filled with sterile silicone, and the animals were left to recover for at least 16 hr. Over the next 1–7 days, we performed repeated-insertion recordings using either NeuroNexus or Neuropixels silicon probes. For a subset of animals, we dipped the probe into DiI on the last day of recording, and perfused the animal to perform post-hoc tracing of the electrode recording locations. For the remaining recordings, the probe position was determined using anatomical landmarks (i.e., bregma and lambda).

NeuroNexus recordings were performed using either a Tucker-Davis Technologies digitizer and custom-written MATLAB code as described previously (*Ahmadlou and Heimel, 2015*). Neuropixels recordings were performed using a National Instruments I/O PXIe-6341 module and SpikeGLX (https://github.com/billkarsh/SpikeGLX). Visual stimulation was performed as described previously (*Montijn et al., 2016a*), and synchronized with high accuracy ( < 1 ms) using photodiode signals that recorded visual stimulus onsets. Spikes were sorted post-hoc using Kilosort2 (https://github.com/MouseLand/Kilosort2, *Pachitariu, 2021*) and only clusters of sufficient quality, as defined by Kilosort2's default threshold, were included for further analysis. High-quality clusters (i.e. putative neurons) were assigned a brain region using the AllenCCF MATLAB toolbox (https://github.com/cortex-lab/allenCCF, *Peters, 2021*), which automatically calculates a neuron's anatomical position based on penetration location, angle, and depth of the silicon probe. Abbreviations for brain areas mostly follow the Allen Brain Atlas area codes. V1: primary visual cortex. AM: anteromedial visual cortex. PM: posteromedial visual cortex. LGN: lateral geniculate nucleus. LP: lateral posterior nucleus. NOT: nucleus of the optic tract. APN: anterior pretectal nucleus. SC: superior colliculus. All code used in laminar probe data acquisition and pre-processing is available online (https://github.com/Jorrit-Montijn/Acquipix, *Montijn, 2021*).

## Visual stimulus parameters

Visual stimuli during Neuronexus and Neuropixels recordings were shown at 60 Hz on a 51 by 29 cm Dell screen at 17 cm distance from the animal's eyes, using Psychtoolbox three in Matlab. Drifting gratings were displayed within a 120 visual-degree diameter window with two visual-degree cosine edge that faded smoothly into a neutral-gray background. Drifting gratings were shown in 24 directions: [0, 15, … 345] degrees at a spatial frequency of 0.05 cycles per degree and a temporal frequency of 1 cycle per second. Natural movies were 20 s long and consisted of four distinct scenes taken from the BBC nature documentary Earthflight (*Montijn et al., 2016a*).

## Acquisition and preprocessing of retinal multi-electrode array data

Adult ( > 1 year) zebrafish (*Danio rerio*), were dark-adapted for at least 1 hr. Then under IR illumination fish were euthanized by rapid immersion in ice-cold water, the eyes removed, the retina isolated and placed photoreceptor side up on a perforated 60 electrode array (60pMEA200/30iR-Ti using a MEA2100 system: Multichannel systems, Reutlingen, Germany) in a recording chamber mounted on an Nikon Optiphot-2 upright microscope and viewed under IR with an Olympus 2 x objective and video camera (Abus TVCC 20530). Room temperature Ames' medium (Sigma-Aldrich) gassed with a mixture of $O_2$ and $CO_2$ at pH of 7.6 continuously superfused the MEA recording chamber.

Extracellular multiunit GC activity was recorded at 25 kHz in MC rack (Multichannel systems, Reutlingen, Germany), zero-phase bandpass filtered (250–6250 Hz) with a fourth-order Butterworth filter in Matlab (MathWorks, Natick, MA, USA), and sorted into single-unit activity with 'offline spike sorter'

(Plexon, Dallas, TX, USA). Spikes were detected using an amplitude threshold $> 4\sigma_n$ where $\sigma_n$ is an estimation of the background noise

$$\sigma_n = median\left(\frac{|x|}{0.6745}\right) \tag{44}$$

with $x$ being the bandpass-filtered signal (*Quiroga et al., 2004*). The detected spikes were manually sorted into single units based on the first two principal components versus time.

The light stimulus consisted of a 500 ms full field light flash, preceded and followed by a 500- and 1000 ms period of darkness, generated using Psychophysics Toolbox Version 3 (*Brainard, 1997*; *Kleiner et al., 2007*), and repeated either 50 or 100 times. Stimuli were projected onto the retina from the photoreceptor side by a DLP projector (Light Crafter 4500, Wintech, Carlsbad, CA, USA) using a custom-built 2 x water immersion objective. Only white light stimuli were used. The "dark" light intensity was 6 μW/m2, and the maximal 'light' intensity was 176.2 μW/m$^2$.

## Acquisition and preprocessing of calcium imaging data

Drifting grating responses were recorded as previously described (*Montijn et al., 2016a*) and putative spike times were extracted using an exponential fitting algorithm (*Montijn et al., 2016b*). All codes used in pre-processing of drifting grating calcium imaging data are available online (https://github.com/JorritMontijn/Preprocessing_Toolbox).

Virtual corridor experiments were performed on adult male RCFL-tdTOM x PV FlpO x VIP2r-cre (n = 1) and VIP2r-cre mice (n = 3). During water restriction, the weight of the animals was carefully monitored and kept stable through powdered milk intake (10 % diluted in drinking water) in the corridor or additionally fed solid drink. Mice were kept on a reversed day/night schedule (12 hr/12 hr) and experiments were performed in their active dark phase.

For virus injection and chronic window implantation procedures, animals were anesthetized with isoflurane at 5 % for induction and 1.0–1.5% for maintenance. Metacam (1 mg/kg) was administered subcutaneously for systemic analgesia and dexamethasone (8 mg/kg) was administered to prevent brain swelling. The eyes were covered throughout the surgeries with Cavasan ointment. For virus injections, three small holes were drilled around the center of the right V1 (2.9 mm lateral from midline, 0.5 mm anterior from lambda). Animals were injected with saline-diluted AAV2/1.hSyn. GCaMP6f.WPRE.SV40 (Addgene) at a depth of 250 and 550 μm (final concentration, $10^{12}$ viral particles/ml; 36.8 nl/depth/location). Mice were allowed to recover for a minimum of 2 weeks in their home cage before window implantations. For chronic window implantations, a custom-made head bar was positioned above V1 and fixed on the skull with dental cement (Kerr). A circular 3 mm craniotomy was made on the area of injection while the dura was kept intact. A double glass window (3 + 4 mm diameter) was placed inside the craniotomy and fixed on the skull with dental cement (Kerr). Animals were allowed to recover for a minimum of 1 week before training began.

Before imaging, mice were trained to being head fixed and run through a virtual corridor on a custom-made treadmill (*Figure 7A*). Running speed of the mice was measured with a rotary encoder and processed using an Arduino and Matlab. Absolute running speed was used to render the virtual corridor in real-time. The left half of the virtual corridor was rendered on a gamma-corrected monitor (Dell) placed under an angle of 45° with a mirror to create the perception of a symmetrical corridor. The virtual corridor was written in Matlab using OpenGL and Psychophysics Toolbox three and contained a 100 cm black and white Gaussian noise texture with overlying visual stimuli. We used three vertical gratings and three checkerboard stimuli positioned 11 cm apart, at locations 22–77 cm (*Figure 7B*). After completing a run in the corridor, mice were immediately shown a luminance-matched gray screen followed by a 0.5 s auditory cue (8 kHz) 1 second later. Two seconds after this cue, the mice received a 5–10 μl milk reward. Mice were trained for ~ 10 sessions until they completed up to ~ 150 trials during imaging sessions.

Imaging was performed with a two-photon microscope (Neurolabware) equipped with a Ti-sapphire laser (Mai-Tai 'Deepsee', Spectraphysics; wavelength, 920 nm) and a 16 x, 0.8 NA water immersion objective (Nikon). The microscope was controlled by Scanbox (Neurolabware) running on Matlab. One one mouse we performed dual-plane imaging at 15.5 Hz/plane using an electrically tunable lens (OptoTune). The other three mice were imaged in a single plane at 15.5 Hz. During the pre-processing stage, we discarded all interneurons and only included putative pyramidal cells for further analysis.

## Analysis of virtual corridor joint-feature encoding

For each neuron, we calculated ZETA-scores after aligning the putative spike times to trial start ('time-aligned'), converting spike times to locations and aligning them to the beginning of the corridor ('location-aligned'), or aligning the spike times to a visuomotor mismatch event, where rendering of the virtual corridor was paused for 500 ms ('mismatch-aligned') (*Figure 7C*). We call the resulting values time-modulation, location-modulation and mismatch-modulation respectively. In the case of the spatial "location-aligned" analysis, we furthermore discarded the first and last 10 % of the track to avoid including the start and end box locations. We investigated whether there was a relationship between the modulation values that neurons showed for these three different features in two ways. First, we simply calculated the Pearson correlation at the level of a single recording between time/location, time/mismatch, and location/mismatch modulations (*Figure 7D–F*). A one-sample t-test on these correlation values showed there was a significant, positive correlation for time/location and a significant, negative correlation for location/mismatch. However, this analysis does not directly answer the question whether the joint encoding of two features at the level of single neurons is more or less likely than we would expect by chance.

We therefore z-scored the modulation values per recording and pooled all neurons. We then used a kernel-density estimator (KDE) to construct a probability density estimate for the distribution of modulation values for time, location, and mismatch (*Figure 7—figure supplement 1*). Combining these distributions for a pair of features, we obtain a null-hypothesis distribution of what the joint-feature-encoding distribution would look like if the modulation-scores are independent of each other. Using the same bandwidths as for the single-feature KDE-distributions, we smoothed the real data so we could directly compare which regions are over- or under-represented in the real data, producing the heat maps in *Figure 7G–I*. To quantify this over/underrepresentation, we counted the number of neurons in the upper-right quadrant, where neurons lie that have high modulation scores for both features. We compared the real count to the distribution obtained from 10,000 random samples taken from the joint-feature KDE-derived null-hypothesis distribution, where there is no correlation between the two feature modulation scores. As shown in *Figure 7G-I*, *Figure 7—figure supplement 1*, time and location-modulation scores were more often both high in single neurons than expected from chance. Moreover, there were fewer neurons that showed a joint encoding of both spatial location and visuomotor mismatch than expected from chance.

## VIP-optogenetics analysis of Allen Brain Institute cephys data

Detailed information on experimental and data acquisition procedures can be found online at the Allen Brain Institute website: https://portal.brain-map.org/explore/circuits/visual-coding-neuropixels. We used data from 5 Vip-IRES-Cre; Ai32 mice that underwent laser-based optogenetic stimulation. We pre-selected 1706 clusters that were recorded in visually-responsive cortex (AL, AM, PM, L, V1, RL, or MMP) and were of sufficient quality, specifically: KiloSort2 tagged the cluster as "good", ISI violations were under 0.5, amplitude thresholds under 0.1, and presence ratios over 0.9. The next step was to cull this population to only cells that showed modulated spiking with respect to the onset of optogenetic stimulation. We therefore included N = 1144 cells that showed $P < 0.05$ with a ZETA-test on the window between (–0.5, + 0.5 s) after optogenetic stimulation. We calculated the peak-response latency using our IFR method and discarded cells with peak responses earlier than 1 ms after optogenetic stimulation onset (N = 909). The remaining cells were classified as VIP if their IFR peak latency was < 10 ms (N = 13), as Inhibited if their firing rate was significantly lower during 10–30 ms after optogenetic stimulation than during the 50 ms preceding optogenetic onsets (N = 59), as Activated if their rate was significantly higher during 10–30 ms after optogenetic onset than during the pre-stimulus baseline (N = 137), and otherwise as Other (N = 700). The large size of this latter group could be explained by many cells being visually-responsive to the blue laser light. We chose a 10–30 ms window to compare the IFR peak/trough latencies of Activated and Inhibited cells, as 10 ms was the time of laser offset, and visual responses start to emerge in visual cortex after about 30–50 ms.

## A Proof of Time-Invariance

We have described some properties of the ZETA-test in the main method section, but we have not yet explained what the function is of the mean-subtraction of $\delta$ in *Equation 14*. This step plays a critical

role in ensuring that the ZETA-test is time-invariant: i.e., that the latency of a neuronal response with respect to the stimulus onset does not affect the statistical significance of the ZETA-test.

We can see that this is the case if we have made a specific choice for the trial onsets, consisting of consecutive intervals of $\tau$, and made a set of $n$ spike times $v_1$ to $v_n$ in the interval $[0, \tau]$. First we rewrite *Equations 1–4* as:

$$\delta_i = \frac{i}{n} - \frac{v_i}{\tau}$$

(45)

Recall that $d_i = \delta_i - 1/n \sum_i \delta_i$ (*Equations 6*; *7*). Now, consider a shift of the trial onset times by $\Delta$ and let $v_k$ be the highest spike time smaller than $\Delta$. This results in a new set of $n$ spike times $v_i'$:

$$v_i' = v_{i+k} - \Delta \quad \text{for} \quad 1 \leq i \leq n - k$$

(46)

$$v_i' = v_{i+k-n} - \Delta + \tau \quad \text{for} \quad n - k + 1 \leq i \leq n$$

(47)

Note that *Equation 47* implies circular time with the recording wrapping back to the beginning at the end of all trials, which we assume here to keep $n$ constant. If we define $\delta_i'$, analogous to $\delta_i$, and use *Equation 46–47* we find:

$$\delta_i' = \delta_{i+k} - \frac{k}{n} + \frac{\Delta}{\tau} \quad \text{for} \quad 1 \leq i \leq n - k$$

(48)

$$\delta_i' = \frac{i}{n} - \frac{v_{i+k-n} - \Delta + \tau}{\tau}$$

$$= \delta_{i+k-n} - \frac{k}{n} + \frac{\Delta}{\tau} \quad \text{for} \quad n - k + 1 \leq i \leq n$$

(49)

And if we subtract its mean, the constants are removed, and we get:

$$d_i' = d_{i+k} \quad \text{for} \quad 1 \leq i \leq n - k$$

(50)

$$d_i' = d_{i+k-n} \quad \text{for} \quad n - k + 1 \leq i \leq n$$

(51)

The set of $d_i'$ are thus identical to the set $d_i$ except for a reordering. The maximum of the set $|d_i|$ will therefore also be the maximum of the set $|d_i'|$. This means that it does not matter when the trial onsets are taken to compute ZETA. Note that this is not the case if $\delta_i$ is used instead of $d_i$.

To illustrate this difference, we will now derive closed-form solutions for the expectation and variance of $\delta_i$ and $d_i$ in the specific case of step-wise changing Poisson-distributed spiking rates. Note that this section serves only to illustrate the above derivation (*Equation 45–51*) for a specific case, so the reader may choose to skip the rest of this section without missing out on any particularly important comments.

First, recall the base variables: we use **v** as a vector of spike times relative to stimulus onset, with the total onset-to-onset epoch duration defined as $\tau$ (see *equation 1*). We will set the neuron's firing probability to be homogeneous with Poisson rate $\lambda$. Since exponentially distributed inter-spike intervals generate Poisson-distributed spike counts, we use:

$$S \sim \text{Exp}\left(\frac{1}{\lambda}\right)$$

(52)

Therefore, spike time $w_i$ is:

$$w_i = w_{i-1} + S_i$$

(53)

In the limit of large $n$, the incremental deviation $\dot{D}_i$ from an exactly uniform spike-time distribution (i.e., $1/\lambda$) is therefore:

$$\dot{D}_i = \frac{1}{\lambda} - S_i$$

(54)

In reality, however, the rate $\lambda$ is estimated from data that are limited by the observation window $\tau$, number of trials $m$, and observed number of spikes $n = m\tau\lambda$. First we collapse all spikes over trials,

following *Equation 1*, such that $v_n = \tau$. Then we normalize by $\tau$ to make $v_n = 1$, which means that we now have:

$$\sigma \sim \text{Exp}\left(\frac{1}{m\tau\hat{\lambda}}\right) = \text{Exp}\left(\frac{1}{n}\right) \tag{55}$$

generating spike times

$$v_i = v_{i-1} + \sigma_i \tag{56}$$

with incremental deviations

$$\dot{\delta}_i = \frac{1}{m\tau\hat{\lambda}} - \sigma_i \tag{57}$$

We can confirm this is correct by rewriting the total deviation $\delta_i$ at spike $i$ as:

$$
\begin{aligned}
\delta_i &= \sum_{j=1}^{i}\left(\dot{\delta}_j\right) \\
&= \frac{i}{m\tau\hat{\lambda}} - v_i \\
&= \frac{i}{n} - v_i
\end{aligned}
\tag{58}
$$

To compute a closed-form for the variance of $d$, we need the first and second central moments of $\delta$ (i.e., the mean and variance). $\delta$ depends on $\dot{\delta}$, the variance of which is:

$$
\begin{aligned}
Var\left[\dot{\delta}\right] &= Var\left[\text{Exp}\left(\frac{1}{n}\right)\right] \\
&= \frac{1}{n^2}
\end{aligned}
\tag{59}
$$

In the case for large $n$, each $\dot{\delta}$ is an exponential random variable. As $\delta_i$ is a sum over $\dot{\delta}$s, we can approximate its pdf as an Erlang distribution with scale parameter $k$ equal to the spike number $i$:

$$
\begin{aligned}
\Delta_i &= \sum_{j=1}^{i}\left(\dot{\delta}_j\right) \\
\Delta_i &\sim \text{Erlang}\left(k = i, \lambda = \frac{1}{n}\right)
\end{aligned}
\tag{60}
$$

The Erlang distribution is a special case of the gamma distribution with discrete parameters. An equivalent formulation is therefore:

$$\Delta_i \sim \Gamma\left(i, \lambda\right) \tag{61}$$

However, we have so far ignored that $\delta_i$ is mean-zero and fixed at 0 at t = 0 and t = $\tau$. For a Wiener process W, such fixed points are known as a Brownian bridge (*Mansuy and Yor, 2008*), which is described by:

$$B\left(t\right) = W\left(t\right) - \frac{t}{\tau}W\left(\tau\right) \tag{62}$$

However, since the underlying stochastic process in our case is not a standard normal, but gamma-distributed, we cannot directly apply the above equation. Instead, we found that with sufficiently large $n$, the behavior of $\delta$ is described by a weighted difference of two time-symmetric series of gammas (see *Figure 2—figure supplement 1B*). One series grows as *Equation 60*, whereas its symmetric counterpart shrinks as $k = n\text{-}i$:

$$\delta_i = \frac{n-i}{2n}\Gamma\left(i, \lambda\right) - \frac{i}{2n}\Gamma\left(n - i, \lambda\right) \tag{63}$$

The central moments of the difference between two gamma distributions are given by *Klar, 2015*:

$$\mu = \frac{\alpha_1}{\beta_1} - \frac{\alpha_2}{\beta_2} \tag{64}$$

$$\sigma^2 = \frac{\alpha_1}{\beta_1^2} + \frac{\alpha_2}{\beta_2^2} \tag{65}$$

Where the subscripts indicate the two distributions and $\alpha_1 = k = i$, $\alpha_2 = n - i$, and $\beta_1 = \beta_2 = 1/\lambda$. Filling in the above parameters and weighting variables, we therefore get:

$$E\left[\delta_i\right] = \frac{n-i}{2n}i\lambda - \frac{i}{2n}\left(n-i\right)\lambda = 0 \tag{66}$$

$$\begin{aligned} Var\left[\delta_i\right] &= \frac{n-i}{2n}i\lambda^2 + \frac{i}{2n}\left(n-i\right)\lambda^2 \\ &= \frac{ni\lambda^2 - i^2\lambda^2}{n} \end{aligned} \tag{67}$$

Now, we need to compute the variance over $d_i$, which is defined as the mean-subtracted $\delta_i$.

To simplify the following derivation, we will assume that time is circular (as in *Equation 47*) and that jittering does not strongly impact the number of spikes $n$, so we can we treat $n$ as a fixed number of samples. As shown in *Figure 2—figure supplement 1C*, these assumptions allow accurate estimations. Now, we consider $v_k$, $1 \leq k \leq n$ taken from a uniform distribution on [0,1] and ordered such that $v_1 \leq v_2 \leq \ldots \leq v_n$. We recognize the $v_k$ as the order statistics of the sample, for which the probability distribution is

$$p\left(v_k\right) = \frac{n!}{(n-k)!(k-1)!}v_k^{k-1}\left(1-v_k\right)^{n-k} \tag{68}$$

The expectation of $v_k$ and $v_k^2$ are given by

$$E\left(v_k\right) = \frac{n!}{(n-k)!(k-1)!}\int_0^1 dv_k v_k^k \left(1-v_k\right)^{n-k} = \frac{k}{n+1} \tag{69}$$

$$E\left(v_k^2\right) = \frac{n!}{(n-k)!(k-1)!}\int_0^1 dv_k v_k^{k+1} \left(1-v_k\right)^{n-k} = \frac{k\left(k+1\right)}{(n+2)(n+1)} \tag{70}$$

Now, redefine

$$\delta_i = \frac{i}{n+1} - v_i \tag{71}$$

Then using *Equation 69* and *Equation 70*, we find

$$E\left(\delta_i\right) = 0 \tag{72}$$

$$E\left(\delta_i^2\right) = \frac{i(n-i+1)}{(n+1)^2(n+2)} \tag{73}$$

We find that the variance of $\delta_i$ is parabolic with respect to i, with its minimum at $i = 1$ and $i = n$ and its maxium at the middle between those points. The maximum of all $\delta_i$ is thus also more likely to at the middle $\delta_i$ than at the extremes.

We define again

$$d_i = \delta_i - \bar{\delta} \tag{74}$$

with $\bar{\delta} = 1/n\Sigma_i\delta_i$. We know already that $E(d_i^2)$ is the same for all i, and we can thus write

$$E\left(d_i^2\right) = \frac{1}{n}\sum_k E\left(d_k^2\right) = \frac{1}{n}\sum_k E\left(\delta_k^2\right) - E\left(\bar{\delta}^2\right) \tag{75}$$

To compute $E(\bar{\delta}^2)$, we need to know $E\left(\delta_i\delta_j\right)$ and therefore $E\left(v_iv_j\right)$. To compute $E\left(v_iv_j\right)$ for $j > i$, we can write $v_j = v_i + x_{j-i}$ and understand that $x_{j-i}$ follows the same order statistic distribution as $v_i$ except that there are now only $n - i$ samples taken from an interval $[0, 1 - v_i]$ and $x_{j-i}$ is the j-i$^{th}$ sample.

$$E\left(v_iv_j\right) = E\left(v_i\left(v_i + x_{j-i}\right)\right) = \frac{i(j+1)}{(n+1)(n+2)} \tag{76}$$

Using this we see that for $j > i$

$$E\left(\delta_i\delta_j\right) = E\left(\left(\frac{i}{n+1} - v_i\right)\left(\frac{j}{n+1} - v_j\right)\right) = \frac{i(n-j+1)}{(n+1)^2(n+2)} \tag{77}$$

Then we find, after a long but straightforward calculation, that

$$E\left(\bar{\delta}^2\right) = \frac{1}{n^2}\sum_i\sum_j E\left(\delta_i\delta_j\right) = \frac{1}{n^2}\sum_{i=1}^{n}\left\{\sum_{j=1}^{i-1}E\left(\delta_i\delta_j\right) + \sum_{j=i}^{n}E\left(\delta_i\delta_j\right)\right\} = \frac{1}{12n} \tag{78}$$

And therefore

$$E\left(d_i^2\right) = \frac{1}{n}\sum_k E\left(\delta_k^2\right) - E\left(\bar{\delta}^2\right) = \frac{1}{6(n+1)} - \frac{1}{12n} = \frac{n-1}{12(n+1)n} \tag{79}$$

Note that the dependence on *i* has disappeared: i.e., the variance of *d* is time-invariant; which is what we aimed to show. Also note that while we made various assumptions to simplify the above derivations, our theoretical solutions accurately predict simulated data (*Figure 2—figure supplement 1*); showing these assumptions have little impact on the results.

If we could now compute Var[max($d'$)], E[max($d'$)] and E[max($d$)] from the above solutions for $d'$, we would even be able to construct a closed-form solution for the ZETA-test's p-value in the case of exponentially-distributed inter-spike intervals. Unfortunately, the distribution of max($d$) is unknown and fairly complex, because the elements of *d* are not statistically independent.

## Acknowledgements

Brain images in *Figures 3–5* were generated using Brain Explorer 2 (*Lau et al., 2008*). We thank the Allen Brain Institute for their openly accessible data sets. We also thank the engineers of the mechatronics workshop at the NIN.

## Additional information

### Funding

| Funder | Grant reference number | Author |
| --- | --- | --- |
| Stichting Vrienden van het Herseninstituut | | J Alexander Heimel |

The funders had no role in study design, data collection and interpretation, or the decision to submit the work for publication.

### Author contributions

Jorrit S Montijn, Conceptualization, Data curation, Formal analysis, Investigation, Methodology, Software, Validation, Visualization, Writing – original draft, Writing – review and editing; Koen Seignette, Marcus H Howlett, J Leonie Cazemier, Data curation, Investigation, Writing – review and editing; Maarten Kamermans, Christiaan N Levelt, Supervision, Writing – review and editing; J Alexander Heimel, Formal analysis, Funding acquisition, Methodology, Project administration, Software, Supervision, Writing – original draft, Writing – review and editing

### Author ORCIDs

Jorrit S Montijn (iD) http://orcid.org/0000-0002-5621-090X
Koen Seignette (iD) http://orcid.org/0000-0002-7398-6291
Marcus H Howlett (iD) http://orcid.org/0000-0001-9620-8014
J Leonie Cazemier (iD) http://orcid.org/0000-0003-2875-6283
Maarten Kamermans (iD) http://orcid.org/0000-0003-0847-828X
Christiaan N Levelt (iD) http://orcid.org/0000-0002-1813-6243
J Alexander Heimel (iD) http://orcid.org/0000-0002-5291-4184

### Ethics

All experiments were approved by the animal ethics committee of the Royal Netherlands Academy of Arts and Sciences, in compliance with all relevant ethical regulations. Animals received anesthetics and analgesics where applicable, such as during surgeries, and every effort was made to minimize animal suffering.

### Decision letter and Author response

Decision letter https://doi.org/10.7554/eLife.71969.sa1
Author response https://doi.org/10.7554/eLife.71969.sa2

## Additional files

### Supplementary files
• Transparent reporting form

### Data availability
As stated in the manuscript, open-source code for the ZETA-test is available at https://github.com/JorritMontijn/ZETA and https://github.com/JorritMontijn/zetapy. Furthermore, code to reproduce the ZETA benchmarks are available at https://github.com/JorritMontijn/ZETA_analysis_repository (copy archived at https://archive.softwareheritage.org/swh:1:rev:58dc4d8d3e9db6c06906445a8c3fa4a253b1fe3a) The Neuropixels data are annotated and available here: https://doi.org/10.5061/dryad.6djh9w108.

The following dataset was generated:

| Author(s) | Year | Dataset title | Dataset URL | Database and Identifier |
|---|---|---|---|---|
| Montijn JS | 2020 | ZETA benchmarking neuropixels data | https://doi.org/10.5061/dryad.6djh9w108 | Dryad Digital Repository, 10.5061/dryad.6djh9w108 |

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
