## [Decision Letter]

**Acceptance summary:**

A central question in neuroscience is whether a neuron responds to a certain stimulus. However, when the response is complex, e.g., bi-phasic where an increase in rate is followed by a lower rate, this is not always easy to determine. In this methods paper, the authors introduce a new bin-less method that detects whether a neuron responds to a stimulus. In particular for labs using high-throughput data collection, the method should be useful.

**Decision letter after peer review:**

[Editors’ note: the authors submitted for reconsideration following the decision after peer review. What follows is the decision letter after the first round of review.]

Thank you for submitting your work entitled "A parameter-free statistical test that improves the detection of neuronal responsiveness" for consideration by *eLife*. Your article has been reviewed by 3 peer reviewers, one of whom is a member of our Board of Reviewing Editors, and the evaluation has been overseen by a Senior Editor. The following individual involved in review of your submission has agreed to reveal their identity: Jonathan D Victor (Reviewer #3).

Our decision has been reached after consultation between the reviewers. Based on these discussions and the individual reviews below, we regret to inform you that your work will not be considered further for publication in *eLife*.

While the reviewers saw considerable merit in the study, in particular regarding the extensive and robust result of applying the method to data, the claims regarding the method being state of the art were found to be insufficiently convincing for e*Life*.

*Reviewer #1:*

This paper introduces a, as far as I know, new method to detect modulation of neural firing rates. It is particularly more sensitive to neurons with a bi-phasic modulation of rate.

The method is straightforward and perhaps not very advanced but should be of interest to a wide community, also because it does not depend on binning.

The study appears to be competently carried out and applies the method to a very large range of systems.

*Reviewer #2:*

This paper will be of interest to researchers using electrophysiology or Calcium imaging who want to detect whether cells are responsive to a stimulus and are concerned that testing the mean firing rate may miss cells that are responsive through temporal patterns of their spike trains. The method reported here is rigorous and is an advance compared to doing a standard statistical test on mean firing rate, since it is sensitive to additional temporal information. However, other methods for detecting responsiveness based on temporal information are available and it is not clear that the new test has a meaningful performance advantage compared to these.

The goal here is (1) to develop a reliable method for detecting whether and when a neuron responds to stimulation and does so without requiring the bin size to be specified and (2) that this method is superior to previous approaches. As stated by the authors "we present the parameter-free ZETA-test, which outperforms t-tests and ANOVAs by including more cells at a similar false-positive rate".

The authors have achieved the first goal; developing a parameter-free "zeta test", which works by testing whether the cumulative distribution of spike times deviates significantly from what would be expected from random firing. However, achievement of the second goal is not proven.

Readers should consider the following when reading the paper:

1. The binless aspect of the zeta test is elegant as it avoids the need to choose a binsize for the analysis.

2. That the zeta test is more sensitive than a t-test is a straightforward consequence of the fact that the two tests are applied to different inputs. The t-test is used to compare whether the mean firing rate in some temporal interval is greater during the stimulus compared to baseline. The zeta test works on the basis of the spike times within the response interval. The increased sensitivity of the zeta test compared to the t-test stems from the property that the timing of spikes within an interval necessarily conveys as much or more information then the mean firing rate within the interval.

3. Given the previous point, a better benchmark for the zeta test is against other approaches that take spike timing into account. The authors consider an ANOVA, which splits the response interval into bins, and tests whether or not the pattern of firing rate over those bins is temporally modulated. They find that the zeta test is more sensitive than the ANOVA for some binsizes. However, for the most physiologically relevant binsizes (those around the time-scale of the stimulus), the two methods perform similarly. The apparent advantage of the zeta test is largest at small bin sizes < 1 ms. It is possible that the apparent performance advantage in these small bins compared to 20-100 ms bins, is a sampling artefact. Further work is needed to exclude this. In sum, contrary to the claim of the abstract, it is not clear that the zeta method has a genuine performance improvement compared to ANOVA.

Overall, following on from the comments above, this method will be of potential interest to a broad community of researchers seeking that measure neural activity with cellular resolution (extracellular electrophysiology and Calcium imaging) but the advance is at a level more suited to a specialised journal.

The authors find the inclusion rates of the zeta and ANOVA to be indistinguishable for binsizes of ~2-20 times the sampling interval of the movie, so within this broad range of timescales, there is no performance advantage for the zeta test. There is an advantage for smaller bins and for larger bins. The advantage for larger bins is presumably for the same reason as occurs in the t-test section (this should be explained in Results). The advantage for small bins is problematic for two reasons. First, the range of bin sizes is taken down to < 0.1 ms. This is 100 times less than the natural time-scale of the movie stimulus (its frame interval) so, although not impossible, it is surprising/interesting/unconvincing that there is extra temporal information there missed by the ANOVA. Another possibility (as extensively considered in the information theory literature, e.g. Panzeri et al., 2007) is that this is a sampling artefact. Further work is needed to address this point, for example by using simulated data where the response time-scales are controlled. It is also possible that the ANOVA is underperforming due to over-conservative nature of Bonferroni correction. the ANOVA's inclusion rate might be better with multiple comparisons control done via the Benjamini-Hochberg-Yekutieli procedure.

*Reviewer #3:*

This work presents a new approach to determining whether neuronal responses are modulated in response to a stimulus. The approach is motivated by the need for statistical tests that do not require a choice of a timeframe for analysis, or other analysis parameters. The investigators derive a null distribution for the new measure, provide code for its implementation, and benchmark it against the t-test for both simulated and real data in several diverse systems.

While I am generally sympathetic to the motivation of the work, I have both conceptual and technical concerns. The most critical is that, while I think they have presented strong evidence that the tool is useful, I don't think that they have shown that it advances the state of the art (point 1 below): they compare their approach to a t-test, which I think is something of a straw man. I believe the biological results are strong and interesting, but, since the paper is submitted as "Tools and Resources" paper, my review focuses on the tool itself, and whether it has been shown to be an advance.

1. The main comparisons are to the t-test. While the t-test may well be widely used and simple, it is far from the state of the art. Whenever a response is modulated in time (and not just steadily elevated), there is almost always a better approach. For example, for periodic stimulation, such as the examples of Figure 2, one can use Fourier analysis to identify modulations in the response. More generally, point process models – see for example the work of Emery Brown and Rob Kass, including the Kass, Eden, and Brown textbook "Analysis of Neural Data", provide a theoretical framework for detecting response modulations in a wide variety of settings, as well as determining when the peak modulation occurs; these methods provide estimates of the underlying firing rate along with error bars. As is the case for the proposed method, these point-process approaches work directly with spike times, and do not require the choice of a bin width. So, while it is a good point that t-statistics may underestimate the fraction of neurons that are modulated by an experimental manipulation in some situations, there are a number of standard solutions already available. The manuscript does not discuss these other approaches or compare the new approach to them, and therefore stops substantially short of showing that the ZETA-test is an advance on the state of the art. [On a subjective note, I think that as a general approach, the point-process framework mentioned above is superior conceptually, as it begins with an explicit statistical model and an inference framework.]

2. For the examples in which the response is constructed to be a pure elevation of the firing rate, the ZETA test does not appear to be superior (Figure 4, row 1): there is a larger fraction of cells detected as responsive, but also a greater number of false positives. An ROC curve would likely show this. But more importantly, if the underlying firing statistics are Poisson – as they are constructed to be – then, since the firing rate is a sufficient statistic, simple estimation of the firing rate cannot be worse than the zeta statistic.

3. The derivation of the null distribution for the statistic makes a lot of assumptions, including the extent of jittering and the use of extreme-value asymptotics; there may be a much simpler way to the goal. If I understand correctly, the idea is to construct an estimate of the cumulative distribution of spike times, and to compare that with the null hypothesis that they are uniformly distributed. If that is the case, perhaps one could just use the Kolmogorov-Smirnov test? This also brings up another issue: if the neuron's underlying firing pattern is far from Poisson, e.g., that it is similar to a renewal process whose interspike interval distribution is highly peaked – will the zeta test work?

4. Finally, I think that the virtues of a "nonparametric" test is not as obvious or universal as they might first appear. On the one hand, even with a test such as the one proposed here, one still needs to choose the time period to analyze. On the other hand, one often cares very much about the timescale of the response variation, both to exclude phenomena that one is not interested in, and to understand what aspects of the response are informative. This point is of course not germane to whether the paper makes an advance, but I think it would be preferable to discuss these considerations, rather than to assert that nonparametric tests are preferable.

---

## [Author Response]

[Editors’ note: the authors resubmitted a revised version of the paper for consideration. What follows is the authors’ response to the first round of review.]

Reviewer #1:This paper introduces a, as far as I know, new method to detect modulation of neural firing rates. It is particularly more sensitive to neurons with a bi-phasic modulation of rate.The method is straightforward and perhaps not very advanced but should be of interest to a wide community, also because it does not depend on binning.

We would like to thank the reviewer for the positive comments and review. As noted above, we now show that the ZETA-test is not only superior in the case of bi-phasic neurons, but also for bursting cells (Figure 4). We have also added an analysis of the marginal case of purely exponentially distributed inter-spike intervals (ISIs); in this hypothetical scenario, the t-test exceeds the ZETA-test.

However, we also show that only small deviations from exact alignment are required for the ZETA test to exceed the t-test’s performance (Figure 4).

The study appears to be competently carried out and applies the method to a very large range of systems. All my comments are minor.Reviewer #2:This paper will be of interest to researchers using electrophysiology or Calcium imaging who want to detect whether cells are responsive to a stimulus and are concerned that testing the mean firing rate may miss cells that are responsive through temporal patterns of their spike trains. The method reported here is rigorous and is an advance compared to doing a standard statistical test on mean firing rate, since it is sensitive to additional temporal information. However, other methods for detecting responsiveness based on temporal information are available and it is not clear that the new test has a meaningful performance advantage compared to these.The goal here is (1) to develop a reliable method for detecting whether and when a neuron responds to stimulation and does so without requiring the bin size to be specified and (2) that this method is superior to previous approaches. As stated by the authors "we present the parameter-free ZETA-test, which outperforms t-tests and ANOVAs by including more cells at a similar false-positive rate".The authors have achieved the first goal; developing a parameter-free "zeta test", which works by testing whether the cumulative distribution of spike times deviates significantly from what would be expected from random firing. However, achievement of the second goal is not proven.

We thank the reviewer for these useful comments on our manuscript. We appreciate that our previous version lacked sufficient comparisons with more advanced statistical approaches. As noted above, we have added a considerable amount of new material to the manuscript, including multiple alternative statistical tests. We hope that the reviewer finds the new comparisons sufficiently convincing. Regarding the comparison with the multi-timescale ANOVA, we believe that our previous manuscript was not sufficiently clear, and we have now rewritten parts of the description of the ANOVA-based analysis.

Readers should consider the following when reading the paper:1. The binless aspect of the zeta test is elegant as it avoids the need to choose a binsize for the analysis.

Thank you, we appreciate the comment.

2. That the zeta test is more sensitive than a t-test is a straightforward consequence of the fact that the two tests are applied to different inputs. The t-test is used to compare whether the mean firing rate in some temporal interval is greater during the stimulus compared to baseline. The zeta test works on the basis of the spike times within the response interval. The increased sensitivity of the zeta test compared to the t-test stems from the property that the timing of spikes within an interval necessarily conveys as much or more information then the mean firing rate within the interval.

We agree with the reviewer that the number of data points used by a t-test (n=number of trials) is much lower than with the ZETA-test (n=number of spikes). However, the t-test does have access to information that the ZETA-test does not: the spike times used by the ZETA-test are flattened over trials, while the t-test uses the variability across trials. Therefore, when the variability across trials is low, but the variability within a trial is high, the ZETA-could perform worse than a t-test. The important analysis here would therefore be to find where their point of equality lies. As we show in our new Figure 4A-B, the t-test exceeds the ZETA-test’s sensitivity only for biologically implausible exponentially-spiking cells. Importantly, even in the case of strongly bursting cells (Figure 4C-F), the ZETA-test outperforms the t-test. We discuss this issue in the paragraph “ZETA-test in the absence of short peaks of activity” (lines 172-202).

3. Given the previous point, a better benchmark for the zeta test is against other approaches that take spike timing into account.

We have now added a detailed comparison of the ZETA-test with alternative tests derived from renewal-process theory to the manuscript (lines 112-117, 542-697, Supplementary Methods and Figure 3 – Supplements 2,3), and performed a new comparison with an optimally-binned ANOVA (lines 149-170, 533-540, Figure 3). The ZETA-test is superior to all alternatives we have tested, and we now also describe more clearly which mathematical properties are crucial to the ZETA-test’s superior statistical sensitivity.

The authors consider an ANOVA, which splits the response interval into bins, and tests whether or not the pattern of firing rate over those bins is temporally modulated. They find that the zeta test is more sensitive than the ANOVA for some binsizes. However, for the most physiologically relevant binsizes (those around the time-scale of the stimulus), the two methods perform similarly. The apparent advantage of the zeta test is largest at small bin sizes < 1 ms. It is possible that the apparent performance advantage in these small bins compared to 20-100 ms bins, is a sampling artefact. Further work is needed to exclude this. In sum, contrary to the claim of the abstract, it is not clear that the zeta method has a genuine performance improvement compared to ANOVA.Overall, following on from the comments above, this method will be of potential interest to a broad community of researchers seeking that measure neural activity with cellular resolution (extracellular electrophysiology and Calcium imaging) but the advance is at a level more suited to a specialised journal.The authors find the inclusion rates of the zeta and ANOVA to be indistinguishable for binsizes of ~2-20 times the sampling interval of the movie, so within this broad range of timescales, there is no performance advantage for the zeta test. There is an advantage for smaller bins and for larger bins. The advantage for larger bins is presumably for the same reason as occurs in the t-test section (this should be explained in Results). The advantage for small bins is problematic for two reasons. First, the range of bin sizes is taken down to < 0.1 ms. This is 100 times less than the natural time-scale of the movie stimulus (its frame interval) so, although not impossible, it is surprising/interesting/unconvincing that there is extra temporal information there missed by the ANOVA. Another possibility (as extensively considered in the information theory literature, eg Panzeri et al., 2007) is that this is a sampling artefact. Further work is needed to address this point, for example by using simulated data where the response time-scales are controlled.

We apologize for not describing this procedure sufficiently clearly. The ZETA-test is intrinsically timescale-free, and we chose different bin widths (i.e., timescales) only for the various ANOVAs. The fact that the ANOVAs perform worse than the ZETA-test at short and long timescales is not an indication that the ANOVA missed information at those timescales. Rather, no information about the stimulus exists at those timescales, and therefore the ANOVA performs poorly. We agree with the reviewer’s logic if this were what our results showed, but this is not the case: the reviewer’s interpretation was caused by us describing our methods insufficiently clearly. We have made various changes to the text (lines 204-229) and the legend of Figure 5 to further clarify that only the ANOVAs used different bin sizes.

It is also possible that the ANOVA is underperforming due to over-conservative nature of Bonferroni correction. the ANOVA's inclusion rate might be better with multiple comparisons control done via the Benjamini-Hochberg-Yekutieli procedure.

Using a less conservative correction procedure would increase the ANOVA’s inclusion, but also increase the number of false alarms. As such, it would not change the ROC curve shown in Figure 5C, except that the curve’s inclusion and FA rates would correspond to different values of the significance level α.

Reviewer #3:This work presents a new approach to determining whether neuronal responses are modulated in response to a stimulus. The approach is motivated by the need for statistical tests that do not require a choice of a timeframe for analysis, or other analysis parameters. The investigators derive a null distribution for the new measure, provide code for its implementation, and benchmark it against the t-test for both simulated and real data in several diverse systems.While I am generally sympathetic to the motivation of the work, I have both conceptual and technical concerns. The most critical is that, while I think they have presented strong evidence that the tool is useful, I don't think that they have shown that it advances the state of the art (point 1 below): they compare their approach to a t-test, which I think is something of a straw man. I believe the biological results are strong and interesting, but, since the paper is submitted as "Tools and Resources" paper, my review focuses on the tool itself, and whether it has been shown to be an advance.1. The main comparisons are to the t-test. While the t-test may well be widely used and simple, it is far from the state of the art. Whenever a response is modulated in time (and not just steadily elevated), there is almost always a better approach. For example, for periodic stimulation, such as the examples of Figure 2, one can use Fourier analysis to identify modulations in the response. More generally, point process models – see for example the work of Emery Brown and Rob Kass, including the Kass, Eden, and Brown textbook "Analysis of Neural Data", provide a theoretical framework for detecting response modulations in a wide variety of settings, as well as determining when the peak modulation occurs; these methods provide estimates of the underlying firing rate along with error bars. As is the case for the proposed method, these point-process approaches work directly with spike times, and do not require the choice of a bin width. So, while it is a good point that t-statistics may underestimate the fraction of neurons that are modulated by an experimental manipulation in some situations, there are a number of standard solutions already available. The manuscript does not discuss these other approaches or compare the new approach to them, and therefore stops substantially short of showing that the ZETA-test is an advance on the state of the art. [On a subjective note, I think that as a general approach, the point-process framework mentioned above is superior conceptually, as it begins with an explicit statistical model and an inference framework.]

We understand the reviewer’s point of view that (A) the ZETA-test was not well described in terms of existing mathematical frameworks and (B) that more specialized and advanced techniques may outperform the ZETA-test.

Regarding point A: In this revised version, we have dedicated significant work to better ground the ZETA-test in existing mathematical frameworks. We compare the ZETA-test to alternative formulations derived from renewal and point-process models (lines 608-697 and Figure 3 – Supplements 2,3), showing how ZETA can be constructed step-by-step from a simple Kolmogorov Smirnov test of real spike-times versus a Poisson H0 distribution up to the full-fledged ZETA-test. Comparing the performance of these various alternative tests, we found that the ZETA-test outperforms all others (Figure 3 – Supplement 3). Moreover, we now describe how the ZETA-test achieves time-invariance (Supplementary Methods), and show that an ANOVA that uses an optimal bin width still performs worse than the ZETA-test (lines 149-170, Figure 3).

Regarding point B: We created the ZETA-test as an easy-to-use, first-pass test for neuronal responsiveness that requires no hyperparameter tuning and is relatively fast, as no such test yet exists for neurophysiological analysis. We do not claim that the ZETA-test always exceeds the performance of more sophisticated and specialized model-based approaches, nor was this our aim. Rather, the point we wish to make in our manuscript is that the ZETA-test is a powerful, statistically sensitive test that exceeds the performance of other naïve approaches. That said, we acknowledge that the range of alternative statistical tests in our previous version was limited. We have therefore added a comparison with a multiplicative inhomogeneous Markov interval (MIMI) model, as described in (Kass and Ventura, 2001; *Neural computation*; Kass, Eden and Brown, 2014; *Analysis of Neural Data*), to the current version. We use it to both detect responsive cells (Figure 3 – Supplement 2) and to determine response latencies (Figure 6). In both cases, the MIMI-method performed somewhat mediocrely. It was very well able to detect responsive cells with many spikes, but showed a high false alarm rate for cells with few spikes, presumably due to overfitting. As we also note in the manuscript, the MIMI-method may be a useful technique in many regards, but it requires hyperparameter tuning, such as regularization and the choice of number and location of knots, in order to perform well. It is therefore not suited as an automated generalist test for responsiveness of neurons with varied spiking statistics. Moreover, any alternative method that uses model fitting will likely show similar requirements. Using the MIMI-method to determine latencies, we found that it showed more timescale-invariance than PSTH-based approaches, but still did not perform as well as the ZETA-IFR (Figure 6). We made numerous edits throughout the manuscript to better communicate the aim and applicability of the ZETA-test, and expanded the discussion of the proposed scope of the ZETA-test (lines 377-418). In summary, full MIMI-model based methods do not seem to be suited for unsupervised, large-scale use as neuronal responsiveness tests.

2. For the examples in which the response is constructed to be a pure elevation of the firing rate, the ZETA test does not appear to be superior (Figure 4, row 1): there is a larger fraction of cells detected as responsive, but also a greater number of false positives. An ROC curve would likely show this. But more importantly, if the underlying firing statistics are Poisson – as they are constructed to be – then, since the firing rate is a sufficient statistic, simple estimation of the firing rate cannot be worse than the zeta statistic.

We have added a more in-depth analysis of how the ZETA-test and t-test behave in the case of cells with purely exponentially-distributed inter-spike intervals (Figure 4A-B). Our analyses show that there indeed exists a range of a parameters where the t-test outperforms the ZETA-test. Importantly, however, this is a fairly narrow range and only holds for biologically implausible parameter values.

3. The derivation of the null distribution for the statistic makes a lot of assumptions, including the extent of jittering and the use of extreme-value asymptotics; there may be a much simpler way to the goal.

As we now also better explain in the methods (lines 514-521), we use the extreme value asymptotics only to limit the number of random samples needed to calculate the p-value of ZETA. In the limit of infinite random jitters, we would obtain the true null distribution. The use of the Gumbel approximation is therefore simply to make the test more computationally tractable.

If I understand correctly, the idea is to construct an estimate of the cumulative distribution of spike times, and to compare that with the null hypothesis that they are uniformly distributed. If that is the case, perhaps one could just use the Kolmogorov-Smirnov test?

We agree with the reviewer that this is indeed an interesting idea. We have therefore explored it in more detail. As we show in lines 608-668 and Figure 3 – Supplement 3, the basis of ZETA is not dissimilar to a K-S test, but a K-S test shows a very high false alarm rate. In short, this is presumably caused by the issue that any deviation from non-uniform ISIs will lead to a low p-value; i.e., stimulus non-responsive, but bursting, cells will cause significant p-values.

This also brings up another issue: if the neuron's underlying firing pattern is far from Poisson, e.g., that it is similar to a renewal process whose interspike interval distribution is highly peaked – will the zeta test work?

Again, we agree with the reviewer that this is an important question, and thank the reviewer for bringing this up. We further investigated this issue using simulated bursting cells. Our new analysis show that even in the absence of peaks, but in the presence of bursting and highly peaked ISI distributions, the ZETA-test performs very well, and better than the t-test (Figure 4C-F).

4. Finally, I think that the virtues of a "nonparametric" test is not as obvious or universal as they might first appear. On the one hand, even with a test such as the one proposed here, one still needs to choose the time period to analyze. On the other hand, one often cares very much about the timescale of the response variation, both to exclude phenomena that one is not interested in, and to understand what aspects of the response are informative. This point is of course not germane to whether the paper makes an advance, but I think it would be preferable to discuss these considerations, rather than to assert that nonparametric tests are preferable.

We have now added some additional comments to the discussion on model-based approaches (e.g., lines 386-391, noting that they may attain better performance than the ZETA-test), but require supervision and hyperparameter selection. As we wrote above, we do not wish to claim that the ZETA-test is the *end-all and be-all* of statistical tests: each test must be chosen for a specific use case. We have added various comments relating to the scope of the ZETA-test throughout the manuscript.